# Dynamic and static control of the off-target interactions of antisense oligonucleotides using toehold chemistry

Chisato Terada[1,2], Kaho Oh[1], Ryutaro Tsubaki[1], Bun Chan [3], Nozomi Aibara [4], Kaname Ohyama [5], Masa-Aki Shibata[6], Takehiko Wada[7], Mariko Harada-Shiba [8,9], Asako Yamayoshi [1] & Tsuyoshi Yamamoto [1] ✉

Off-target interactions between antisense oligonucleotides (ASOs) with state-of-the-art modifications and biological components still pose clinical safety liabilities. To mitigate a broad spectrum of off-target interactions and enhance the safety profile of ASO drugs, we here devise a nanoarchitecture named BRace On a THERapeutic aSo (BROTHERS or BRO), which is composed of a standard gapmer ASO paired with a partially complementary peptide nucleic acid (PNA) strand. We show that these non-canonical ASO/PNA hybrids have reduced non-specific protein-binding capacity. The optimization of the structural and thermodynamic characteristics of this duplex system enables the operation of an in vivo toehold-mediated strand displacement (TMSD) reaction, effectively reducing hybridization with RNA off-targets. The optimized BROs dramatically mitigate hepatotoxicity while maintaining the on-target knockdown activity of their parent ASOs in vivo. This technique not only introduces a BRO class of drugs that could have a transformative impact on the extrahepatic delivery of ASOs, but can also help uncover the toxicity mechanism of ASOs.

An interesting feature of antisense oligonucleotides (ASOs) is their unrivaled specificity for their drug targets, underpinned by the precise Watson-Crick base pairing rule. ASOs are programmed to bind to the complementary short stretches of their disease-relevant RNA targets through hydrogen bonding and manipulate their functions via RNase H-dependent or -independent mechanisms[1]. To date, medicinal chemistry efforts in ASO development have identified some sophisticated nucleic acid analogs, backbone modification chemistries, targeted delivery strategies, and their optimal combinations[2–4]. Such chemistries have refined and improved the pharmacodynamic and pharmacokinetic properties of ASOs with the successful launch of several ASOs on the market[5]. However, to compensate for the pursuit of the drug-like properties, many of these chemistries, including phosphorothioate and high affinity sugar modifications, have been revealed to undermine the appealing specificity of ASOs[6]. The increased unintended interactions between ASOs and biological components (off-target interactions) pose clinical safety liability[7,8], and are particularly problematic for ASOs that support RNase H-mediated mechanisms[9,10].

These off-target interactions can be categorized into two types: hybridization-dependent and hybridization-independent. The former

[1]Department of Chemistry of Biofunctional Molecules, Graduate School of Biomedical Sciences, Nagasaki University, Nagasaki, Japan. [2]JSPS Research Fellow (DC1), Japan Society for the Promotion of Science, Tokyo, Japan. [3]Graduate School of Engineering, Nagasaki University, Nagasaki, Japan. [4]Department of Pharmacy Practice, Graduate School of Biomedical Sciences, Nagasaki University, Nagasaki, Japan. [5]Department of Molecular Pathochemistry, Graduate School of Biomedical Sciences, Nagasaki University, Nagasaki, Japan. [6]Department of Anatomy and Cell Biology, Faculty of Medicine, Osaka Medical and Pharmaceutical University, Takatsuki, Japan. [7]Institute of Multidisciplinary Research for Advanced Materials (IMRAM), Tohoku University, Sendai, Miyagi, Japan. [8]Department of Molecular Innovation in Lipidology, National Cerebral and Cardiovascular Center Research Institute, Suita, Japan. [9]Cardiovascular Center, Osaka Medical and Pharmaceutical University, Takatsuki, Japan. ✉e-mail: tsuyoshi.yamamoto@nagasaki-u.ac.jp

arises when ASOs bind to unintended RNAs in a sequence-dependent manner. Conversely, the latter originates from non-specific interactions between ASOs and proteins, facilitated by either nucleotide sequence or phosphorothioate backbone interactions, and occurs irrespective of hybridization. This form of toxicity is distinctly different from the one mediated by the hybridization-dependent mechanism acting on unintended transcripts.

Previous efforts to reduce the ASO class toxicity are essentially attributable to the prevention of either hybridization-dependent or -independent off-target interactions[7,8,11]. These approaches include (1) implementation of the chemical modification of one or more of the ribose[12–16], phosphate[17–20], or base[21] moieties in the ASO strand, (2) removal of the toxic sequence motifs[22,23], and (3) optimization of binding affinity[24–27]. However, although certain beneficial modifications can be identified to mitigate the off-target toxicity inherent to the parent ASO, these modification approaches often yield limited impact or can even exacerbate the toxicity profile[19,21,28]. To date, these proposed methodologies rely on a trial-and-error basis. A more scientifically grounded approach, aimed at concurrently and effectively curtailing non-specific interactions with RNA and proteins, may be preferred in addressing these toxicity challenges. Yet, to the best of our knowledge, no such strategy have been proposed[29,30].

To support this mechanism, we here devise an ASO nanoarchitecture called BRace On a THERapeutic aSo (BROTHERS) or BRO (Fig. 1). BRO is a duplex comprising a typical "gapmer" ASO and a partially complementary peptide nucleic acid (PNA) strand, the latter termed "brother strand or BS" (Fig. 1a, b). The protruding gapmer region of the BRO is termed the "toehold domain," with which the ASO is designed to bind sequence-match RNAs to initiate ASO hybridization. The remaining part is called the "duplex domain." BS acts as a safety apparatus to reduce non-specific interactions by concealing the functional groups of the nucleobases and straightening the phosphorothioate (PS) backbone, both of which are often used as interaction scaffolds by biomolecules[31–34].

Another feature of the technology of the present study is the use of PNA. PNA is an artificial nucleic acid analog that possesses uncharged N-(2-aminoethyl)glycine backbone units to which nucleobases are attached (Fig. 1b)[35–37]. As described below, PNA has some ideal characteristics as BS. PNA exhibits high resistance to both nucleases and proteases, and possesses an unprecedented ability to hybridize with high affinity to both the corresponding antiparallel and aberrant parallel DNA and RNA via Watson-Crick hydrogen bonds to form only half-charged non-canonical duplexes[38]. In addition, PNA per se appears to have markedly smaller non-specific protein interaction properties than ASOs with phosphorothioate (PS) chemistry[39]. Therefore, the non-canonical ASO/PNA hybrids are expected to have reduced protein-binding capacity with ideal thermodynamic stability (Fig. 1c, Mechanism 1).

The BRO system that drives the toehold-mediated strand displacement (TMSD) mechanism can be an architecture that enables ASOs to avoid small mismatches or bulges and reduce hybridization-*dependent* off-target binding (Fig. 1c, Mechanism 2). In these dynamics, the ASO of the BRO duplex screens RNAs to identify their proper targets using its "toehold domain." Thereafter, hybridization proceeds to the duplex domain of the toehold-matched RNAs, potentially replacing the incumbent base pairs of the ASO/BS duplex in a sequential manner. The exploitation of toehold chemistry generally catalyzes and accelerates the DNA strand displacement reaction by ~ $10^6$-fold in solutions[40]. After a temporary back-and-forth branch migration reaction, the back displacement reaction occurs due to the arranged kinetic barrier when a mismatch is detected. Only when ASO does not face mismatches or bulges the branch advances a few bases forward, then the BS spontaneously dissociates from ASO prior to full displacement and the ASO eventually fully binds to the RNA[41]. Although the implementation of this TMSD mechanism would be attractive to improve the target specificity of ASOs, the in vivo operation of such nanodevices using chemically modified DNA has been challenging to date[42].

In this study, we engineer an in vivo operative TMSD system and demonstrate that it enhances the target specificity of conventional

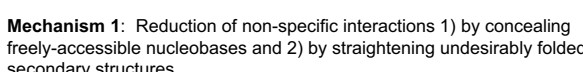

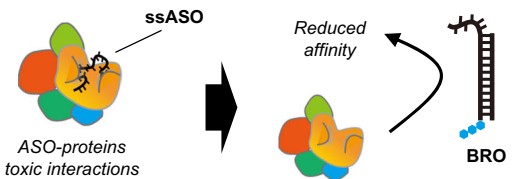

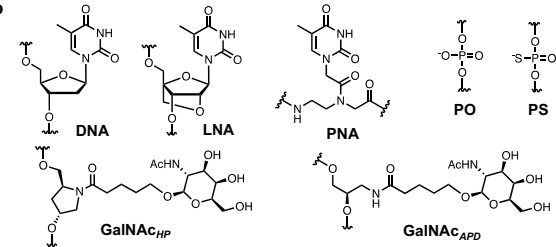

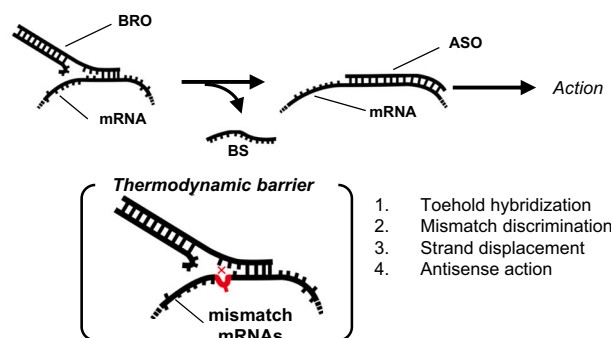

**Fig. 1 | The concept of BROTHERS (BRO) technology and mechanisms of avoiding off-target interactions. a** BRO is an ASO/PNA hybrid, consisting of a toehold domain and a duplex domain. **b** chemistries used in the BROTHERS duplex. **c** putative static (mechanism 1) and dynamic (mechanism 2) mechanisms of action of the BRO duplex.

**Table 1 | Sequence used in this study and melting temperature ($T_m$) of ASO/RNA or ASO/PNA**

| No. | Position ID | Sequence | −2 | −1 | 1 | 2 | 3 | 4 | 5 | 6 | 7 | 8 | 9 | 10 | 11 | 12 | 13 | 14 | 15 | | $T_m$(°C)[a] | Target |
|---|---|---|---|---|---|---|---|---|---|---|---|---|---|---|---|---|---|---|---|---|---|---|
| 1 | mPCS1 | 5′ | Y | Y | A | C | A | c | c | a | a | g | t | t | c | T | C | c | 3′ | | | Pcsk9 (mouse) |
| 2 | mPCS1n | | | | A | C | A | c | c | a | a | g | t | t | c | T | C | c | | | | |
| 3 | mPCS1b | | | b | A | C | A | c | c | a | a | g | t | t | c | T | C | c | | | | |
| 4 | mPCS1f | | | | A | C | A | c | c | a | a | g | t | t | c | T | C | c | f | | | |
| 5 | PNA(12)-mPCS1 | C-term | | | | | pT | pG | pG | pT | pT | pC | pA | pA | pG | pA | pG | pG | N-term | | 79 ± 0.7 | |
| 6 | PNA(11)-mPCS1 | | | | | | | pG | pG | pT | pT | pC | pA | pA | pG | pA | pG | pG | | | 74 ± 0.3 | |
| 7 | PNA(10)-mPCS1 | | | | | | | | pG | pT | pT | pC | pA | pA | pG | pA | pG | pG | | | 63 ± 1.0 | |
| 8 | PNA(C8)-mPCS1 | | | | pT | pG | pT | pG | pG | pT | pT | pC | | | | | | | | | 59 ± 2.1 | |
| 9 | cRNA(14)-mPCS1 | 3′ | | | rU | rG | rU | rG | rG | rU | rU | rC | rA | rA | rG | rA | rG | rG | 5′ | | 62 ± 0.1 | |
| 10 | hApo1 | 5′ | Y | Y | A | A | t | g | g | c | c | a | g | c | T | T | G | | 3′ | | | apoB (human/mouse) |
| 11 | hApo1n | | | | A | A | t | g | g | c | c | a | g | c | T | T | G | | | | | |
| 12 | hApo1b | | | b | A | A | t | g | g | c | c | a | g | c | T | T | G | | | | | |
| 13 | PNA(C5)-hApo1 | C-term | | | pT | pT | pA | pC | pC | | | | | | | | | | N-term | | 42 ± 0.1 | |
| 14 | PNA(C6)-hApo1 | | | | pT | pT | pA | pC | pC | pG | | | | | | | | | | | 43 ± 0.1 | |
| 15 | PNA(C7)-hApo1 | | | | pT | pT | pA | pC | pC | pG | pG | | | | | | | | | | 54 ± 0.1 | |
| 16 | PNA(C8)-hApo1 | | | | pT | pT | pA | pC | pC | pG | pG | pT | | | | | | | | | 58 ± 0.1 | |
| 17 | PNA(C9)-hApo1 | | | | pT | pT | pA | pC | pC | pG | pG | pT | pC | | | | | | | | 59 ± 0.3 | |
| 18 | PNA(C10)-hApo1 | | | | pT | pT | pA | pC | pC | pG | pG | pT | pC | pG | | | | | | | 65 ± 0.3 | |
| 19 | PNA(C11)-hApo1 | | | | pT | pT | pA | pC | pC | pG | pG | pT | pC | pG | pA | | | | | | 69 ± 0.2 | |
| 20 | PNA(C12)-hApo1 | | | | pT | pT | pA | pC | pC | pG | pG | pT | pC | pG | pA | pA | | | | | 78 ± 0.1 | |
| 21 | PNA(C13)-hApo1 | | | | pT | pT | pA | pC | pC | pG | pG | pT | pC | pG | pA | pA | pC | | | | 81 ± 0.1 | |
| 22 | PNA(C9)F-hApo1 | | | F | pT | pT | pA | pC | pC | pG | pG | pT | pC | | | | | | | | | |
| 23 | cRNA(13)-hApo1 | 3′ | | | rU | rU | rA | rC | rC | rG | rG | rU | rC | rG | rA | rA | rC | | | 5′ | 63 ± 0.2 | |

N: LNA, n: DNA, Y: GalNAc$_{APD}$, pN: PNA, rN: RNA, b: biotin, f: Alexa647, F: Cy5

[a] $T_m$ (°C) vs. mPCS1n (2 µM), and vs. hApo1n (4 µM).

ASOs by concurrently mitigating both hybridization-independent and -dependent off-target interactions. We also demonstrate the feasibility of optimizing the pharmacological, toxicological, and pharmacokinetic properties of this system by modulating its structural and thermodynamic characteristics. Our ASO/PNA BRO system could help overcome the long-standing toxicity challenge of ASO therapeutics and enable the elucidation of the toxicity mechanisms that are currently poorly understood.

## Results

### Optimized BROTHERS architectures mitigate hepatotoxicity without affecting antisense activity

A proof-of-concept study was first conducted with mPCS1 ASO (Table 1), a fully PS-modified 14-mer LNA gapmer with two phosphodiester-linked GalNAc$_{APD}$ units, which are the liver-targeting *N*-acetylgalactosamine ligands previously developed by our group[43,44], on its 5′-terminal. mPCS1 targets the murine *Pcsk9* (Proprotein convertase subtilisin kexin 9) mRNA, a well-known hypercholesterolemia target, but does not cover its human ortholog.

Through in-house screening, this ASO was identified to have good knockdown activity; however, it developed dose-dependent moderate hepatotoxicity, typical in the LNA gapmer class of ASOs. At a dose of 100 nmol/kg, mPCS1 achieved 68% knockdown of hepatic *Pcsk9* mRNA in C57BL/6 J mice at 72 h after a single subcutaneous administration; however, a significant increase in the liver transaminase level was observed (Fig. 2a, b). To determine the effect of complementary BS on antisense activity and hepatotoxicity, we prepared four antiparallel-type PNA strands of different lengths and positions: PNA(12), PNA(11), PNA(10), and PNA(C8) (Table 1). As shown in Fig. 2a, the thermodynamic stability of the ASO/PNA BRO duplex, expressed as the melting temperature ($T_m$), increased as the length of BS increased.

Consequently, we observed a consistent decrease in the knockdown activity as the BS length increased. Of note, this trend in activity may not only be due to the stability of the BRO duplex but also the size of the toehold domain.

The effect of BS on the hepatotoxicity of mPCS1 ASO was remarkable, completely suppressing the increase in ALT levels in all treatment groups (Fig. 2b). In particular, mPCS1/PNA(C8) BRO significantly suppressed abnormal ALT elevations to normal levels while maintaining the activity of the parent mPCS1 ssASO. Accordingly, BS screening can yield clinically relevant BROs with superior efficacy and safety. The $T_m$ values were closest to each other for mPCS1/PNA(10) and mPCS1/cRNA(14), whereas that of the mPCS1/PNA(C8) duplex was slightly smaller. This result suggests that the affinity balance between the BRO duplex and ASO/mRNA is of particular importance. The effective doses that reduced the *Pcsk9* mRNA by 50% (ED50) in the liver for mPCS1 and mPCS1/PNA(C8) BRO were 35.3 nmol/kg and 21.7 nmol/kg, respectively (Fig. 2c). As shown in Fig. 2d, dose-dependent ALT elevation was pronounced in the mPCS1-treated arm, while BRO consistently prevented ALT elevation, even at 400 nmol/kg. Thus, the BRO technology can significantly improve the therapeutic margins of ASOs.

To examine the potential hepatotoxicity upon BS dissociation at any later timeframe, we evaluated the antisense activity and toxicity parameters over a longer period after a single 100 nmol/kg administration to C57BL/6 J mice. Ten days of follow-up analysis revealed no predominant difference in knockdown activity between mPCS1 and its BRO counterpart. Further, both systems showed a trend toward the recovery of *Pcsk9* mRNA expression by day 9 (Fig. 2e). In contrast, the mPCS1-treated arm displayed a marked decrease in body weight and the highest ALT levels on day 6, both of which showed a trend toward recovery up to day 9. Throughout this period, there was no change in body weight or abnormal elevation in ALT levels in the BRO-treated arm (Fig. 2f, g).

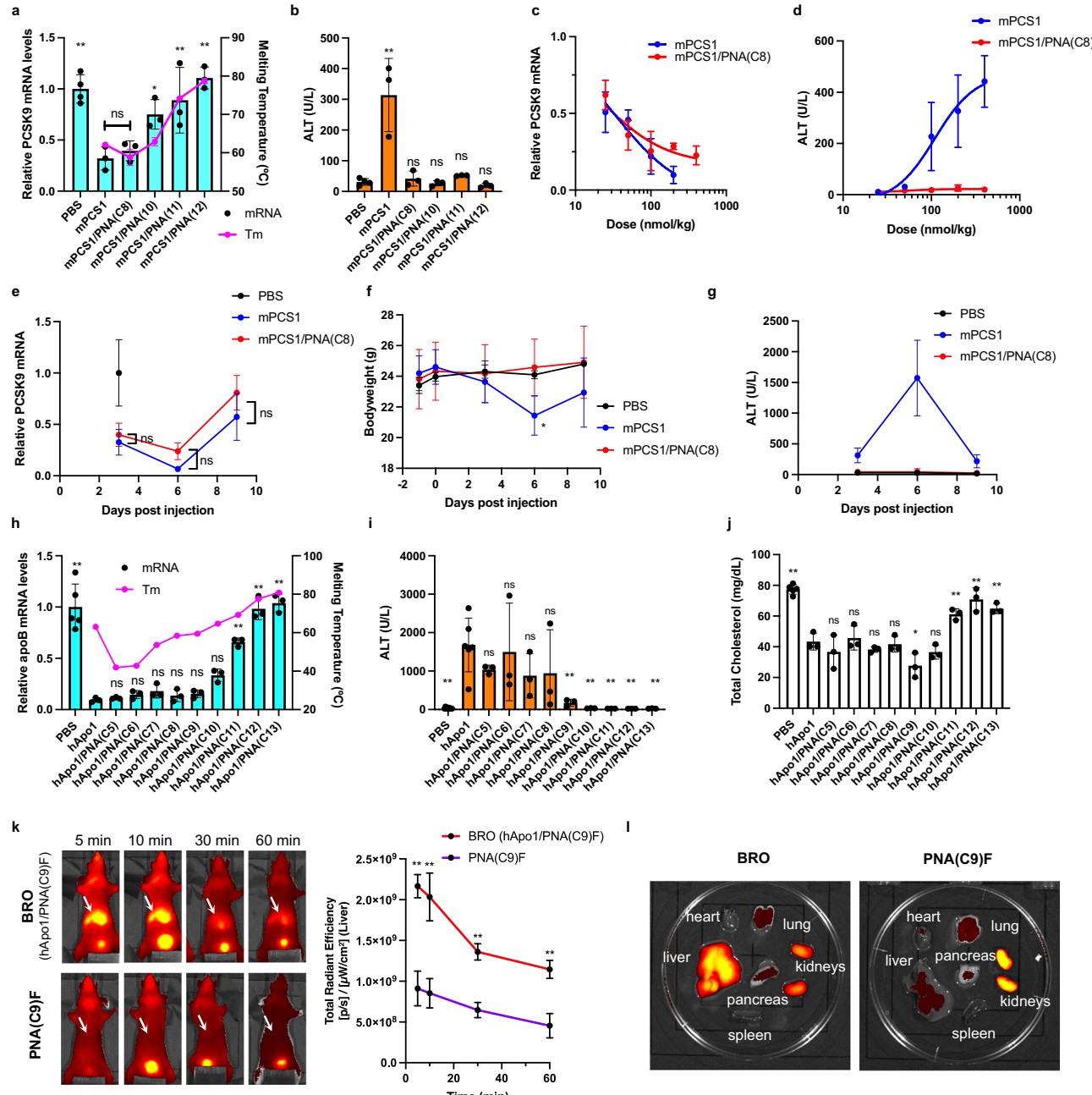

The versatility of the BRO strategy was further investigated by employing another LNA gapmer with a different sequence from mPCS1. This LNA gapmer was hybridized to form an ASO/PNA duplex in combination with a more systematically designed set of BS. Thereafter, structure-activity/toxicity relationship studies were conducted. hApo1 is a 2-8-3 fully PS-modified LNA gapmer that targets a consensus sequence in the human and mouse apolipoprotein B (apoB) mRNA (Table 1). Similar to mPCS1, the hApo1 ASO possesses two phosphodiester-linked GalNAc$_{APD}$ units at the 5′ end. In this study, nine different BS ranging from 5 to 13 bases in length, all covering the 5′ end of the ASO, were tailored.

The $T_m$ values of these nine PNAs versus ASOs ranged from 42 °C to 81 °C, showing a reasonable propensity to increase binding affinity as a function of PNA length (Fig. 2h). The $T_m$ of hApo1/cRNA(13) was 63 °C, which was between that of PNA(9) and PNA(10). To assess the antisense activity and hepatotoxicity of this series of BROs, a single subcutaneous injection of 100 nmol/kg was administered to C57BL/6 J

mice, and the following analyses were performed 72 h post-dosing. Figure 2h shows a comparison of the expression levels of *apoB* mRNA in the liver. Compared to hApo1 ssASO, no statistically significant difference in *apoB* mRNA levels was found for PNA(C5) to PNA(C9)-based BROs, which exhibited high knockdown activity. In contrast, as the strand length increased, a clear decrease in activity was observed from PNA(C10) to PNA(C13). This trend also supports the importance of the binary effects of the binding affinity of BS and the toehold length of BRO, implying the significance of BS detachment from ASO at the appropriate time.

Regarding side effects, hApo1 displayed more severe ALT elevation than mPCS1. BROs ranging from hApo1-PNA(C5) to -PNA(C8) had a limited impact on normalizing ALT levels and could not sufficiently suppress hepatotoxicity. In contrast, BROs longer than PNA(C9), the $T_m$ of which is close to that of the target RNA compared to hApo1, significantly prevented abnormal ALT elevation. Overall, hApo1/PNA(C9) BRO is one of the best combinations that exhibit ALT values in

**Fig. 2 | Optimal brother strands (BS) aid in reducing hepatotoxicity of gapmer ASOs in vivo. a, b** Each mouse (C57BL/6 J, male, *n* = 4 biologically independent samples for control, *n* = 3 for mPCS1 and BRO series) was subjected to a single subcutaneous injection of each *Pcsk9*-targeted drug (ssASO and varied BROs) at a dose of 100 nmol/kg and evaluated 72 h after administration. Hybridization melting temperature ($T_m$) (*n* = 3 independent experiments) and *Pcsk9* mRNA expression (one-way ANOVA followed by two-sided Dunnett's multiple comparison tests comparing to mPCS1, *p* values from left side; 0.0007, 0.9728, 0.0318, 0.0050, 0.0003) (**a**). ALT levels (one-way ANOVA followed by two-sided Dunnett's multiple comparison tests comparing to PBS, *p* values from left side; <0.0001, 0.9737, >0.9999, 0.9982, 0.9999) (**b**). **c, d** Dose dependence of knockdown activity (**c**) and hepatotoxicity (**d**) was evaluated for mPCS1 ssASO and mPCS1/PNA(C8) BRO using the same single-dose experiment in mice (*n* = 4 biologically independent samples for mPCS1 at a dose of 400 nmol/kg, *n* = 3 for other treatment groups). **e–g** Long-term effects of knockdown activity (*n* = 3 biologically independent samples) (**e**), body weight change (*n* = 3 biologically independent samples for control, *n* = 6 for mPCS1 and BRO at day −1, 0 and 3, *n* = 3 for mPCS1 and BRO at day 6; two-way ANOVA followed by two-sided Dunnett's multiple comparison tests comparing to PBS, 0.0283 mPCS1 at Day6) (**f**), and toxicity (*n* = 4 biologically independent samples for control, *n* = 3 for mPCS1 and mPCS1/PNA(C8)) (**g**) after a single administration were compared and evaluated. **h–j** BSOs with various lengths of the BS using ASOs against different target, apolipoprotein B (apoB), were prepared. Their $T_m$ (*n* = 3 independent experiments), *apoB* mRNA knockdown activity (*n* = 5 biologically independent samples for control, *n* = 3 for hApo1 and BRO series; one-way ANOVA followed by two-sided Dunnett's multiple comparison tests comparing to hApo1, *p* values from left side; <0.0001, >0.9999, 0.9966, 0.9188, 0.9992, 0.9909, 0.0729, <0.0001, <0.0001, <0.0001) (**h**), ALT values (*n* = 9 biologically independent samples for control, *n* = 6 for hApo1, *n* = 3 BRO series; one-way ANOVA followed by two-sided Dunnett's multiple comparison tests comparing to hApo1, *p* values from left side; 0.0001, 0.5406, 0.9998, 0.2833, 0.3758, 0.0037, 0.0013, 0.0012, 0.0012, 0.0012) (**i**), and total cholesterol levels (*n* = 5 for control PBS, *n* = 3 for hApo1 and BRO series; one-way ANOVA followed by two-sided Dunnett's multiple comparison tests comparing to hApo1, *p* values from left side; <0.0001, 0.6980, 0.9994, 0.8870, >0.9999, 0.0269, 0.6741, 0.0096, <0.0001, 0.0016) (**j**) were compared with their parent ssASO (hApo1) in mice dosed at 100 nmol/kg at 72 h after injection. **k–l** Pharmacokinetics and biodistribution of fluorescently labeled complementary PNA(C9)F were evaluated under two conditions: in vivo (*n* = 3 biologically independent samples; two-way ANOVA followed by two-sided Bonferroni's multiple comparisons test, *p* values from left side; <0.0001, <0.0001, 0.0004, 0.0006) (**k**) and ex vivo (at 5 min post-dose) (*n* = 1) (**l**); one in the configuration of a BRO duplex and one administered alone. Arrows indicate the liver. All data are presented as mean values ± SD. **$p$ < 0.01, *$p$ < 0.05. "ns" indicates not significant ($p$ > 0.05). Source data are provided as a Source Data file.

---

the normal range while maintaining high knockdown activity. This result was also revealed by the effective reduction in blood cholesterol, which was acutely induced by *apoB* knockdown (Fig. 2j). Taken together, our BROTHERS approach has proven to be a robust approach that can reasonably and easily eliminate the liver toxicity often observed with this class of agents.

## BS circulates alongside ASO in the blood or is otherwise excreted rapidly

To confirm whether the optimized BRO complexes circulated in the blood and were delivered to the target liver as a duplex, an in vivo fluorescent imaging experiment was performed. In particular, C-terminal Cy5-conjugated PNA(C9)F was prepared (Table 1), and its biodistribution was evaluated and compared in vivo when intravenously administered as a double strand (with hApo1) or a single strand. When administered as a BRO duplex, PNA(C9)F mainly accumulated in the liver at the 5-min initial acquisition (Fig. 2k). Subsequently, the fluorescence in the liver slowly faded, as revealed by the 1-h follow-up. This pharmacokinetic property is very similar to that of GalNAc-conjugated PS ASOs, as previously demonstrated by our group[44,45]. On the other hand, PNA(C9)F per se did not exhibit strong fluorescence that spread throughout the body, as observed with BRO, even immediately after administration, but was excreted in urine very quickly. Moreover, fluorescence was hardly observed after 1 h.

In a separate experiment, 5 min after administration, mice were killed, their organs were excised, and the fluorescence intensity in each organ was compared (Fig. 2l). In this ex vivo experiment, PNA(C9)F administered as BRO was found to mainly accumulate in the liver, followed by the kidneys. Single-stranded PNA(C9)F was hardly detected outside the kidneys, implying that PNA itself has a small volume of distribution, which is the characteristic being sought with BS. Overall, these pharmacokinetic/biodistribution analyses revealed that optimized BRO circulates in the blood as a duplex and is transported to the liver very early.

## BRO significantly attenuates changes in gene expression caused by toxic ssASO

We performed RNA sequencing (RNA-Seq) analysis to assess the molecular impact of BS on ASO toxicity. C57BL/6 J mice were subcutaneously treated with 100 nmol/kg of either mPCS1 or mPCS1/PNA(C8) BRO, and global gene expression in the liver was evaluated 72 h post-treatment. We selected day 3 for RNA-seq analysis because,

according to Fig. 2e–g, the toxicity of ssASO was the strongest on day 6, and RNA-seq analysis should be performed at the earlier stages of toxicity. The number of differentially expressed genes (DEGs) was approximately 75% lower in the BRO-treated arm than in the ssASO-treated arm (Fig. 3a). In addition, as depicted in the MA plots, the large deviation in the overall gene expression by mPCS1 ssASO was attenuated in the BRO-treated arm (Fig. 3b, c). To express the total number of mismatches and insertions or deletions between ASO and complementary RNA sequences, we used the value "distance (*d*),"[46] and color-coded the genes showing each *d* value (Supplementary Table 1).

For $d \le 2$, 782 DEGs were detected in the ssASO-treated group, of which 67% exhibited repressed expression. In contrast, only 205 DEGs were observed in BRO, with only 60% being downregulated. For $d > 2$, 49% of the 1361 DEGs in the ssASO group were assigned to suppressed expression, while 38% of the 357 DEGs in the BRO group were classified as suppressed. The number of DEGs and downregulated DEGs was markedly reduced by BRO implementation. The larger fraction of DEGs downregulated for $d \le 2$ compared to $d > 2$ could be due to a hybridization-dependent off-target knockdown; however, the hybridization-independent mechanism may also be involved, as a significant number of genes were upregulated at each *d* value.

A comparison of the gene expression profiles between ssASOs and BRO may reveal how BRO suppresses hepatotoxicity. Thus, we performed gene set enrichment analysis (GSEA) of DEGs. The activation of pathways related to proteasome proteolysis and glutathione (GSH) metabolism was observed (Fig. 3d). In contrast, biochemical reactions that generate reactive oxygen species (ROS), such as drug metabolism pathways, biological oxidative pathways, respiration, and oxidative phosphorylation, were widely attenuated by ssASO treatment (Fig. 3d). To further illustrate the detailed mechanisms of the reduced hepatotoxicity of BRO, we employed weighted parametric gene set analysis (wPGSA) to predict the activity of the transcription factors (TFs) from the transcriptome data obtained[47].

Figure 3e shows a list of TFs suggested to be specifically activated by ssASO rather than BRO and their relative and individual Enrichment Scores. Highly activated TFs include STAT1, STAT2, and IRF3, which are responsible for innate and acquired immune responses and host defenses against infection. IRF1 also regulates the transcription of interferon (IFN) and IFN-inducible genes, which are responsible for immunity, inflammation, cell proliferation and differentiation, and programmed cell death, following DNA damage.

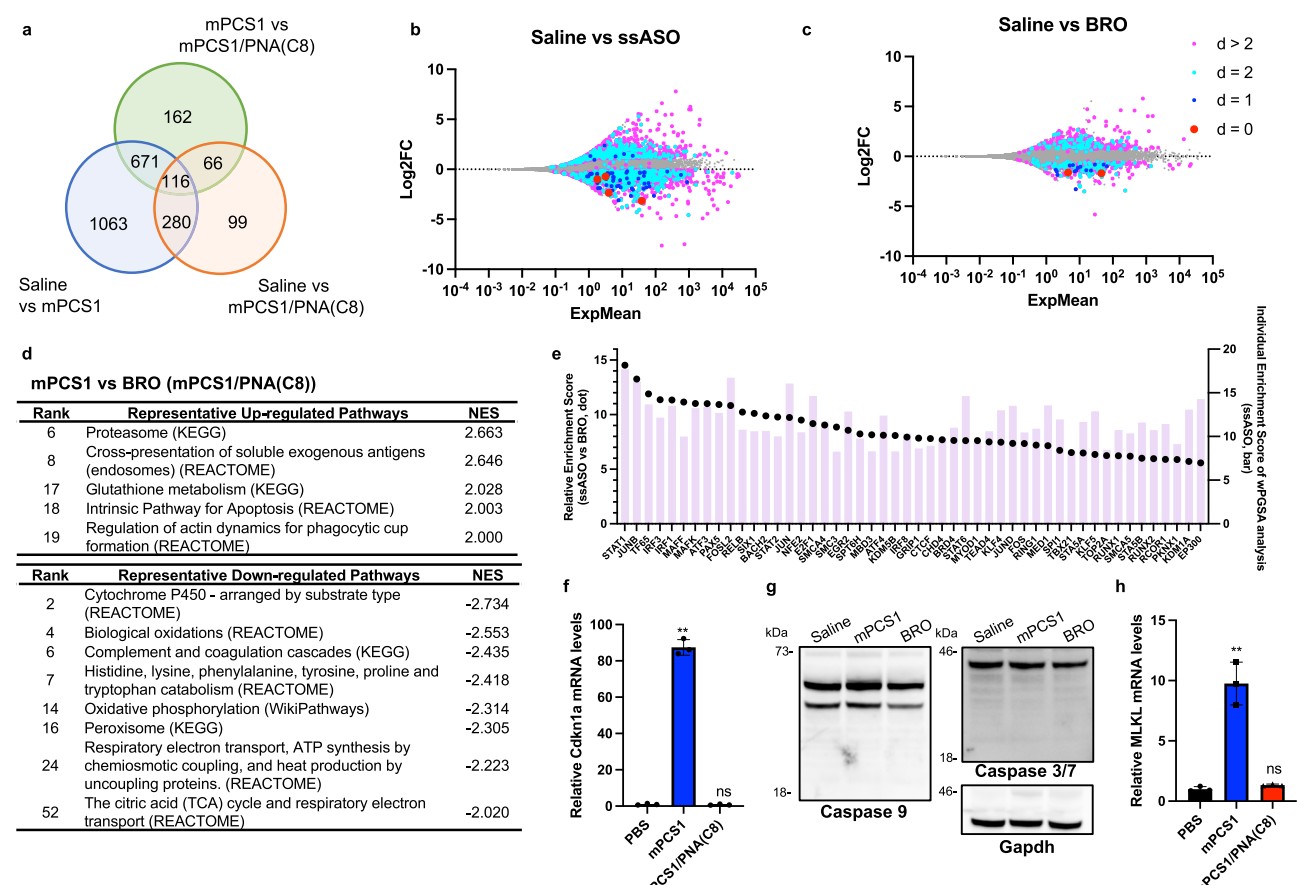

**Fig. 3 | BS installation significantly suppresses toxic ssASO-induced changes in gene expression in vivo. a** RNA-seq analysis of the liver of C57BL/6 J mice treated with either mPCS1 or mPCS1/PNA(C8) BRO subcutaneously at 100 nmol/kg at 72 h after injection. Number of differentially expressed genes (DEGs) in each treatment group (n = 2 biologically independent samples). **b**, **c** MA plots of RNA-seq data with DEGs color-coded according to d-values. **d** GSEA analysis derived from comparison of mPCS1 vs mPCS1/PNA(C8) (Supplementary Data 2). **e** top 50 transcription factors predicted to be particularly active in ssASO over BRO using weighted parametric gene set analysis (wPGSA) analysis based on the RNA-seq data. **f**, **h** The relative mRNA expression of Cdkn1a and Mlkl in the livers of C57BL/6 J mice treated with either mPCS1 or mPCS1/PNA(C8) BRO subcutaneously at 400 nmol/kg at 72 h after injection. (n = 3 biologically independent samples; one-way ANOVA followed by two-sided Dunnett's multiple comparison tests comparing to PBS, Cdkn1a: $p < 0.0001$ mPCS1, $p = 0.9921$ BRO, Mlkl: $p < 0.0001$ mPCS1, $p = 0.8936$ BRO). **g** The western blotting analysis of the caspases in the livers. Shown is a representative result taken from 3 independent experiments. $**p < 0.01$, $*p < 0.05$. "ns" indicates not significant ($p > 0.05$). Data in (**f**, **g**) are presented as mean values ± SD. Source data are provided as a Source Data file.

TF65/Rela and Relb are known to play important roles in inflammation, immunity, differentiation, cell proliferation, and apoptosis. Further, small Mafs are essential for oxidative stress response. In response to DNA damage or other cellular stresses, Cdkn1a/p21 can be induced to arrest the cell cycle for DNA repair or apoptosis if damage cannot be repaired.

The upregulation of Cdkn1a (Fig. 3f) indicated the involvement of apoptosis; however, no detectable activation of caspases related to apoptosis was observed (Fig. 3g). In contrast, the upregulation of Mlkl (mixed lineage kinase domain-like protein) indicates the involvement of necroptosis, a form of programmed cell death that occurs when apoptosis is blocked (Fig. 3h). Mlkl is a protein that plays a critical role in necroptosis[48]. Further research is needed to understand the role of Cdkn1a and Mlkl in regulating the balance between apoptosis and necroptosis. Overall, these results suggest that cellular damage may be the result of immune and/or metabolic responses to foreign invasion and/or intracellular stresses, such as drug-induced perturbations of homeostasis.

## ssASO stimulates cell death pathways, while BRO does not stimulate such pathways

Owing to the in vivo RNA-Seq analysis, we opted to focus on the potential of BRO to suppress innate immune receptor activation and/ or reduce hybridization-independent interactions that generate ROS. ROS are produced by mitochondrial oxidative metabolism and cellular responses to xenobiotics and cytokines. ROS are also intriguing regulators of cell death[49]. To further examine the molecular mechanisms by which BRO selectively removes ssASO toxicity, in vitro evaluation was deemed important. However, a major challenge for the current in vitro systems is the ability to recapitulate and predict the in vivo toxicity of ASOs using common cultured cell lines as a surrogate for cumbersome "in vivo" screening assays[23]. One non-native aspect of culture systems is the high reliance on glycolysis for cellular ATP production rather than the mitochondria, a phenomenon known as the Warburg effect. Under such conditions, the pivotal roles of the mitochondria, such as ROS production and cell death induction, are compromised, and cells are more tolerant against toxicants[50,51]. Supplementation of galactose in the culture medium, instead of glucose, is known to shift the metabolism to mitochondrial ATP production, reproducing in vivo-like conditions. We hypothesized that the ASO toxicity observed in vivo could be evaluated under such particular culture conditions if a common intracellular toxicity mechanism exists.

Previously, we developed a transfection method that better reflects the in vivo activity of ASO, called the calcium-enrichment in culture medium (CEM) method[45,52]. To develop a more sophisticated

transfection method to predict toxicity and activity based on the CEM method, the cells were incubated for 2–5 days in the presence of either 2 μM mPCS1n ssASO or mPCS1n/PNA(C8) BRO in a modified CEM condition (9 mM CaCl$_2$ and 10% FBS with 10 mM galactose, without glucose) (Fig. 4a). Loading ssASO under the modified CEM condition led to the display of cytotoxicity after 2 days. Moreover, after 72 h, the number of cells decreased, and propidium iodide (PI)-positive cells increased (Fig. 4b).

Surprisingly, BRO did not exhibit any cytotoxicity under galactose conditions. Cells in glucose medium were markedly less responsive to ssASO toxicity. Figure 4c compares the concentration-dependent cytotoxicity of ssASO and BRO in galactose medium. The cytotoxicity of ssASO occurred in a concentration-dependent manner; however, the cells survived well when administered any concentration of BRO, which is consistent with the observation in vivo (Fig. 2d). The culture conditions that encouraged greater dependence on the mitochondria yielded this phenotypic deviation, prompting us to speculate that ssASO toxicity is mediated via the mitochondrial pathway. Thus, we evaluated the activity of caspases 8 and 9, which are involved in mitochondria-mediated apoptosis (Fig. 4d). Both caspases were moderately but significantly activated by ssASO treatment in the galactose CEM condition, but were not activated by BRO, remaining unchanged.

Among the more downstream apoptotic caspases, ssASO had a more pronounced activating effect on caspase 3/7 under galactose conditions. In contrast, BRO did not mediate the activation of caspase 3/7 under either condition. The depolarization of the mitochondrial membrane potential (ΔΨm) was also examined using JC−1 dye (Fig. 4e). JC−1 aggregation is associated with high ΔΨm, which is indicated by red fluorescence. ssASO, but not BRO, was found to relax ΔΨm, and red fluorescence was no longer observed. Overall, under the in vivo mimetic conditions, BS installation clearly mitigated the inherent cytotoxicity of ssASO, which could be achieved via direct or indirect effects on the mitochondria.

### Hybridization-independent mechanism: Complementary PNA strand mitigates non-specific protein interactions and changes the intracellular fate of ASO

To determine how bracing ASO with partially complementary PNA can boost safety, we examined the protein-binding property, which is thought to be one of the origins of the hybridization-independent off-target toxicity of ASOs. Biotin-bound mPCS1b ASOs were immobilized on avidin-coated magnetic beads and exposed to murine liver lysate. Thereafter, the bound protein was pulled down (Fig. 5a). After the binding proteins were eluted competitively, they were fractionated via SDS-PAGE and identified using LC/MS. Gel electrophoresis revealed that BRO bound significantly less amount and type of protein than ssASO (Fig. 5b). Similar experiments with another toxic hApo1 ASO revealed a gradual decrease in protein binding as the length of the PNA increased relative to the single strand (Fig. 5c). Interestingly, a marked decrease in toxicity in vivo and reduced protein binding were observed for those longer than PNA(C9).

Most ssASO-binding proteins identified via LC/MS were nucleic acid-binding proteins responsible for homeostasis, cell death, and immune responses (Fig. 5d). To determine the effect of such differences in protein binding characteristics on intracellular dynamics, Alexa647-labeled mPCS1f ASO was transfected into Huh-7 cells as ssASO or the corresponding mPCS1f/PNA(C8) BRO optimized in Fig. 1 using the CEM method, and comparative observations were performed using confocal microscopy. Notably, ssASO displayed a strong signal in the nucleolus, whereas BRO exhibited markedly suppressed localization to the nucleolus (Fig. 5e). This observation suggests that BRO translocates to the cytoplasm as a duplex, and its subcellular localization is altered by binding to proteins that differ from its ssASO counterpart.

### Hybridization-dependent mechanism: complementary PNA strand mitigates off-target hybridizations

Next, we asked if BRO indeed undergoes the thermodynamic and kinetic hybridization mechanism depicted in the Introduction, and if BRO thereby suppresses hybridization-dependent off-target knockdown. A key to directly observe the effect of the toehold on the strand exchange/displacement reaction is to prepare and compare structures incorporating toeholds with those that maintain blunt ends on both sides, ensuring equivalent thermodynamic stability. In this context, we here designed and synthesized two types of BS -cGapmer (LNA3) and cGapmer (LNA4)- by integrating different chemistries (DNA, LNA, and PS) (Table 2). Neither of these variants incorporates a toehold but they exhibit similar melting temperatures ($T_m$) to toehold-bearing PNA(C9).

To compare the susceptibility of these three BROs to the strand exchange reaction, an in-cuvette FRET (fluorescence resonance energy transfer) assay system was devised using TAMRA-labeled hApo1nF and BHQ2-labeled mRNA mimic, ApoB, as shown in Table 2. Specifically, in this system, an equimolar amount of the mRNA mimic was added to each cuvette containing each BRO duplex, and the time-dependent quenching of TAMRA fluorescence was monitored for the efficiency of the strand exchange reaction (Fig. 6a). In conducting in-cuvette FRET experiments with hApo1nF/PNA(C9), hApo1nF/cGapmer (LNA3), and hApo1nF/cGapmer (LNA4), effective fluorescence quenching was observed for hApo1nF/PNA(C9), while largely inefficient quenching was confirmed for both the hApo1nF/cGapmer (LNA3) and hApo1nF/cGapmer (LNA4) systems (Fig. 6b).

These results suggest that the exchange reaction is not merely free energy difference driven, but rather, the kinetic effect of the toehold plays a primary role. Furthermore, when we transfected these BRO duplex pairs into Huh-7 cells by using CEM transfection method, there was an excellent match between the trend of strand exchange rates shown in the FRET assay and the knockdown activity (Fig. 6c). These results provide evidence that the antisense effect of the BROTHERS system is not a simple thermodynamic origin due to a thermodynamic difference (ΔG) between the initial (BRO) and final (ASO/mRNA) duplex states, but that its specially prepared toehold plays a particularly important role in driving the TMSD reaction.

To further investigate BRO's ability to inhibit hybridization-dependent off-target knockdown, we here selected some mRNAs with single mismatches, which can still serve as good substrates of RNase H[24], as representative RNA off-targets. Note also that the evaluation was conducted in vitro as in vivo is more likely to be susceptible to physiological secondary changes. The effects of hApo1n ssASOs and their various BRO counterparts on Huh-7 cells were evaluated using the CEM method (Fig. 6d). The results yielded an ordinal activity sequence that was very similar to the in vivo activity sequence (Fig. 2h). A remarkable decrease in activity was observed between PNA(C9) and PNA(C10) in vitro. A series of van't Hoff analyses of the $T_m$ revealed that the $\Delta G_{310K}$ values of hApo1n/PNA(C8) and /PNA(C9) BROs were slightly lower than those of hApo1n/cRNA(13), while that of hApo1n/PNA(C10) was slightly higher. Such finding indicates the existence of a thermodynamic threshold between PNA(C9) and PNA(C10), which allows the TMSD reaction to occur (Fig. 6e). As shown in Table 2, the $T_m$ values of three mRNAs (Copg, Mast2, and Hltf) containing a single nucleotide mismatch vs. hApo1n were markedly smaller than that of the target ApoB mRNA vs. hApo1n.

To determine the effect of introducing a thermodynamic barrier that disfavors the TMSD reaction on hybridization-dependent off-target knockdown, CEM transfections were performed in Huh-7 cells using hApo1n ssASO and BROs (hApo1n/PNA(C8) and hApo1/PNA(C9)) (Fig. 6f). All agents showed excellent on-target knockdown of ApoB mRNA, with no significant differences between the groups. In contrast, ssASO exhibited significant off-target knockdown of the three single-mismatch-containing genes, while both BROs were found to be free

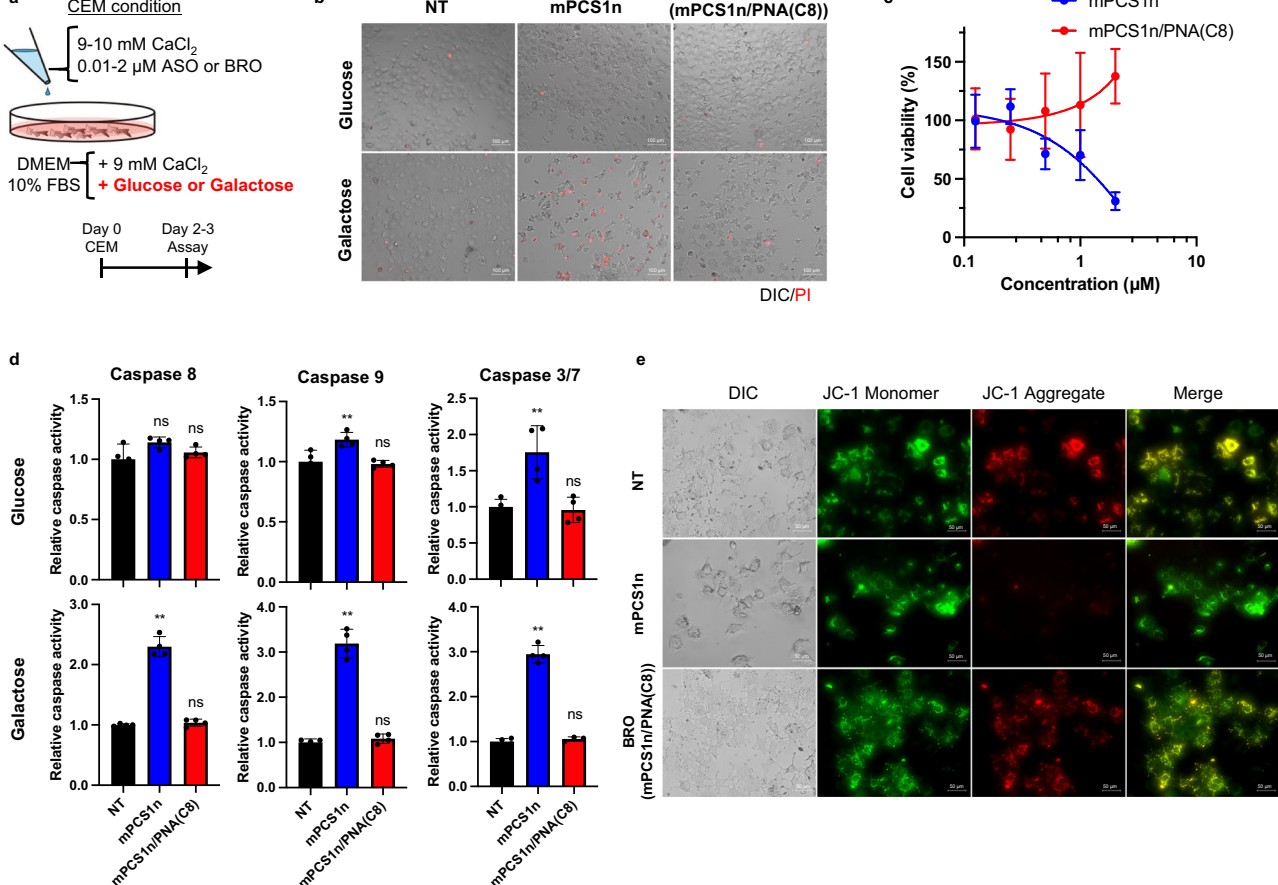

**Fig. 4 | BS installation attenuates mitochondrial damage and cytotoxicity of ssASO in vitro. a** Procedure for evaluation of cytotoxicity of mPCS1n ssASO and mPCS1n/PNA(C8) BRO in original and galactose-modified media. **b** PI staining of Huh-7 cells exposed to 2 μM mPCS1n ssASO and mPCS1n/PNA(C8) BRO after 48 h of CEM transfection ($n = 3$ biologically independent samples). **c** Drug concentration-dependent assessment of cell viability ($n = 5$ biologically independent samples) **d** Activity of caspases 3/7, 8, and 9 ($n = 4$ biologically independent samples; one-way ANOVA followed by two-sided Dunnett's multiple comparison tests comparing to

control, $p$ values from left side; glucose conditions: 0.0674, 0.5360 for caspase 8; 0.0077, 0.9104 for caspase 9; 0.032, 0.9571 for caspase 3/7; galactose conditions: <0.0001, 0.8379 for caspase 8; <0.0001, 0.7712 for caspase 9; <0.0001, 0.7933 for caspase 3/7). **e** the depolarization of the mitochondrial membrane potential (ΔΨm) with JC–1 dye were compared and evaluated for mPCS1n or mPCS1n/PNA(C8) BRO through transfection into Huh-7 cells using the CEM method ($n = 1$) (**e**). **\*\***$p < 0.01$, **\***$p < 0.05$. "ns" indicates not significant ($p > 0.05$). Data in Fig. 4c and d are presented as mean values ± SD. Source data are provided as a Source Data file.

from these off-target effects. To evaluate the susceptibility of BRO to the TMSD reaction for these mismatch genes, the above-described in-cuvette FRET assay system was again utilized.

As shown in Fig. 6g, the FRET assay results revealed that the strand exchange reaction proceeded very efficiently for BRO. Further, binding to *ApoB* mRNA was observed at a rate comparable to that with ssASO. Figure 6h shows that hApo1nF ssASO efficiently binds to all mismatch genes, except Mast2. Of note, the FRET experiment for the Mast2 mRNA mimic was not operative, which might be due to its intrinsic sequence that possibly forms higher-order structures in the strand (Supplementary Fig. 1). In contrast, Fig. 6i shows that the binding rates of BRO to the mismatch genes were markedly slower than that of *ApoB* mRNA. These results suggest that optimizing the thermodynamic stability of BRO itself initiated by the binding of the toehold domain, and the thermodynamic stability of ASO to target and mismatch genes enables effective discrimination of on-target from off-target.

### Parallel binding mode of PNA suitable for longer ASOs

As described above, antiparallel PNAs tend to reach sufficient binding affinity at a length of approximately 10 bases against ASOs. When the BRO strategy was intended to be applied to ASOs with lengths approaching 20 bases, there were concerns that the single-stranded

toehold region would become too long and the design concept of the off-target suppression of the BRO would not be fully realized. In this context, the possibility of using the parallel binding mode of PNA for BS, which tends to have a weaker affinity to complementary strands than antiparallel PNAs[36], was investigated with an 18-mer LNA gapmer introduced by Kasuya et al. [53] as a model. Table 3 lists the sequences used in this study.

Acsl1 ssASO is an 18-mer LNA gapmer targeting a shared sequence in both human and mouse acyl-CoA synthetase long-chain family member 1 (*Acsl1*) mRNA. Considering off-target toxicity, a core focus of this manuscript, longer ASOs generally seem advantageous due to a ostensible reduction in the theoretical number of off-target sequences/sites compared to shorter ones[54]. Indeed, this 18-nt long ASO has a unique sequence with no full match or single-base mismatch/deletion, besides the target *Acsl1* mRNA (Supplementary Table 8). Nevertheless, this ASO has been documented to exhibit hepatotoxicity comparable to the two above-described shorter ASOs. Currently, no comprehensive experimental outcomes unequivocally indicate the superiority of longer ASOs[27,55]. Thus, offering a BRO option applicable to ASOs of such lengths also holds value.

Parallel PNAs are bound by pairing the 5′ end of ASO with the N-terminus of PNA and the 3′ end of ASO with the C-terminus of PNA. Although the detailed binding mode of DNA/RNA vs. parallel PNA has

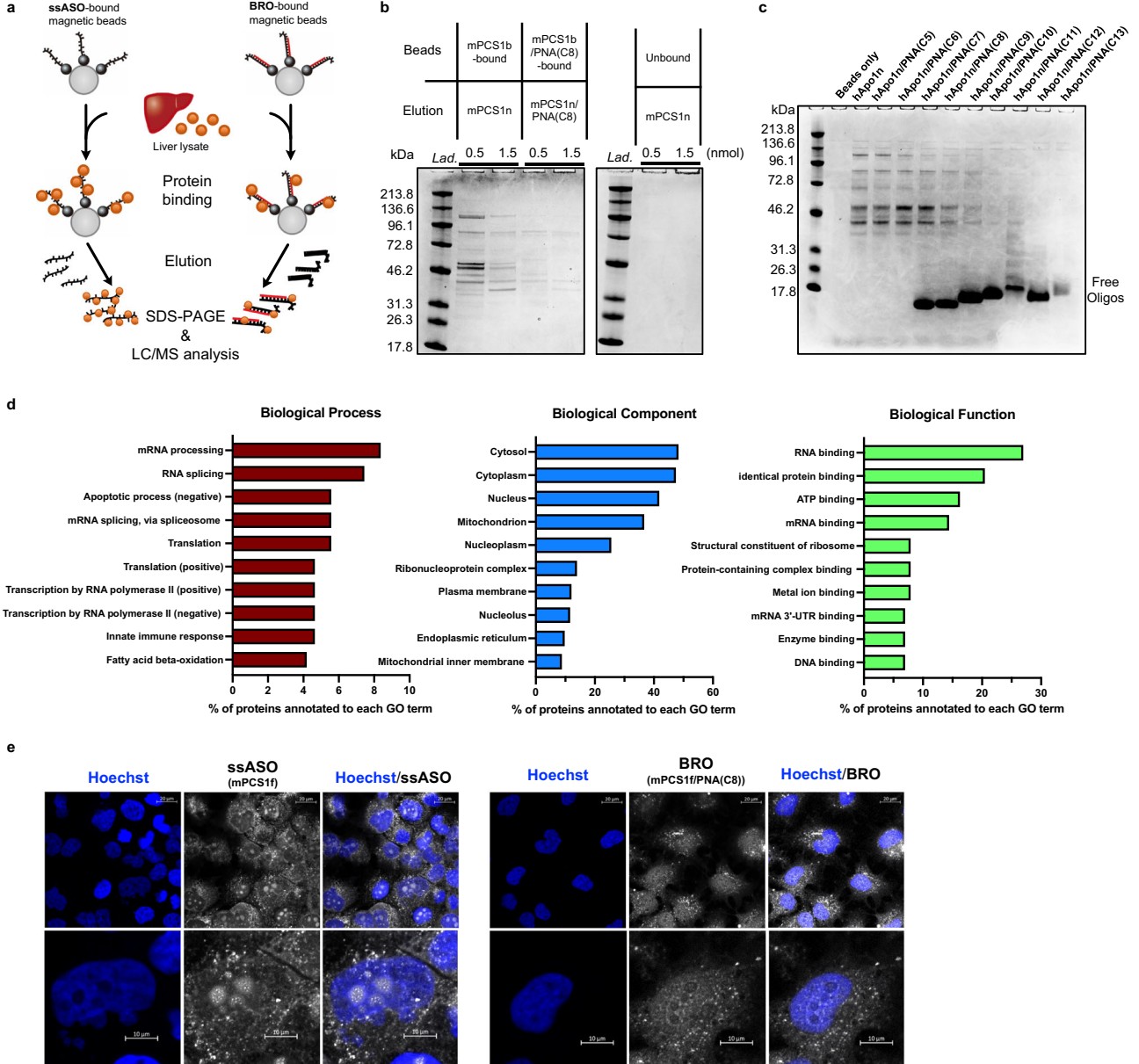

**Fig. 5 | Complementary PNAs change protein-binding property and intracellular fate of ssASO. a** Schematic illustration of competitive protein pull-down assay, where ASO binding proteins are indicated using colored balls, and biotin tags is indicated by the black circles at the ends of ssASO or BRO. **b**, **c** Identification of liver-derived proteins bound by CBB staining for mPCS1b ssASO and mPCS1b/PNA(C8) BRO (*n* = 3 independent experiments) (**b**) and for hApo1b and hApo1b/ PNA(C5–13) BROs (*n* = 2 independent experiments) (**c**). **d** Classification of identified binding proteins using LC/MS proteomic analysis. **e** Intracellular localization of Alexa647-labeled mPCS1f or mPCS1f/PNA(C8) BRO visualized using confocal microscopy (*n* = 2 independent experiments). Source data are provided as a Source Data file.

not been fully revealed, a very well-organized duplex has been depicted via in-house molecular modeling (Fig. 7a), carried out using the density functional tight-binding GFN1-XTB method with the aqueous environment represented by a solvent continuum. Here, parallel PNAs are represented as pPNAs, and five different pPNAs (14–18 nucleotides in length from the C-terminus) were constructed, as shown in Table 3. The optimized structure retains a near-perfect pairing of the bases, with only slight distortions associated with the differences of the PNA backbone from that of the ASO. The melting temperatures of Acsl1n ssASO vs. pPNA and *Acsl1* mRNA are summarized in Fig. 7b and Table 3. A comparison of the $T_m$ values suggested that the binding affinity of Acsl1 ssASO to mRNA falls between pPNA(C16) and pPNA(C17). C57BL/ 6J mice were treated subcutaneously with 150 nmol/kg ssASO or BROs, killed, and dissected after 96 h.

Figure 7c shows a graphical representation of *Acsl1* mRNA expression in the liver and ALT levels in serum. Acsl1 ssASOs displayed high knockdown activity and high ALT levels. BRO attenuated ALT elevation with increasing pPNA length. However, similar to the above results for antiparallel PNAs, the knockdown activity gradually decreased with increasing pPNA length. Among the BS evaluated in this study, pPNA(C15) and pPNA(C16) were considered the most thermodynamically balanced BRO constructs. The increase in liver weight associated with the hepatotoxicity of Acsl1 ssASO was also attenuated by the application of BRO (Fig. 7c). Kasuya et al. reported that Acsl1 ssASO induces cytokines; therefore, we evaluated whether BRO could alleviate this induction. As shown in Fig. 7e, among the cytokines evaluated, BRO suppressed the significant increase in TNFα and IL-4 caused by Acsl1 ssASO. Finally, to compare

**Table 2 | Sequences with single nucleotide mismatches and melting temperature ($T_m$). Blue indicates the mismatch nucleotide**

| Position | | -1 | 1 | 2 | 3 | 4 | 5 | 6 | 7 | 8 | 9 | 10 | 11 | 12 | 13 | | |
|---|---|---|---|---|---|---|---|---|---|---|---|---|---|---|---|---|---|
| Target mRNAs | Sequence | | | | | | | | | | | | | | | | $T_m$ (°C) |
| hApo1nF | 5' | F | A | A | t | g | g | c | c | a | g | c | T | T | G | 3' | |
| PNA(C9)-hApo1 | C-term | | pT | pT | pA | pC | pC | pG | pG | pT | pC | N-term | | | | | 59 ± 0.3 |
| cGapmer(LNA3) | 3' | | T | t | a | c | c | g | g | t | c | g | a | A | C | 5' | 57 ± 0.2 |
| cGapmer(LNA4) | | | T | T | a | c | c | g | g | t | c | g | a | A | C | | 60 ± 0.1 |
| ApoB | | Q | rU | rU | rA | rC | rC | rG | rG | rU | rC | rG | rA | rA | rC | | 63 ± 0.2 |
| Copg | | Q | rU | rU | rA | rC | rC | rG | rG | rG | rC | rG | rA | rA | rC | | 54 ± 0.1 |
| Mast2 | | Q | rU | rU | rA | rC | rC | rG | rG | rU | rG | rG | rA | rA | rC | | 50 ± 0.2 |
| Hltf | | Q | rU | rU | rA | rC | rC | rG | rG | rU | rU | rG | rA | rA | rC | | 51 ± 0.2 |

N: LNA, n: DNA, pN: PNA, F: TAMRA, Q: BHQ2, rN: RNA.

the protein-binding properties of these constructs, the same pull-down assay shown in Fig. 5 was performed (Fig. 7f). Based on the results, protein binding was markedly suppressed at all strand lengths by pPNA compared with single-stranded ASO. The bands indicating proteins tended to become dimers with increasing strand length, and the band intensity was the weakest for pPNA(C16), which had the best balance between activity and toxicity.

## Discussion

BRO was found to successfully disentangle the complex interplay between the activity and toxicity of ASOs in vivo. The three LNA gapmers used in this study represent the currently prevalent gapmer-type ASOs, which have varying lengths, numbers of modifications, and targets. Therefore, the moderate to high hepatotoxicity exhibited by these ASOs may result from both hybridization-dependent and -independent toxicity at different contribution rates. In this study, we have demonstrated that BS contributes to the reduction of these two fundamental off-target interactions that gapmer ASOs generally show.

### Hybridization-independent mechanism

Firstly, our pull-down experiments revealed that the introduction of BS can reduce hybridization-independent off-target (i.e. protein-binding) interactions. Owing to a smaller single-stranded region, BRO has a smaller protein-binding capability (Figs. 5 and 7). Thus, one of the major sources of rich protein-binding capacity can be ascribed to the environmental exposure of functional group-rich nucleobases[56]. In other words, covering functional group-rich nucleobases with complementary strands helps ASO evade protein recognition. Certain sequence motifs, such as tcc, aa, and cc, which were included in the ASOs used in this study, have been reported to contribute to toxicity[22,23].

As described in the introduction, previous studies have demonstrated that the introduction of small structural changes or perturbations at one or a few ribose, phosphate, or nucleobase moieties, or removal of these toxic motifs can regulate non-specific protein binding and reduce the hepatotoxicity of ASOs. However, the impact of such minor modifications on toxicity is limited to addressing the clinical toxicity challenge. Thus, shielding nucleobases or toxic motifs by utilizing complementary strands is a more rational approach to reduce hybridization-independent interactions. In the same context, the lack of charge and ribose as well as poor protein-binding nature of PNA also gives ASOs non-canonical characteristics that differ from single- or double-stranded DNA/RNA, possibly playing a role in further reducing off-target protein binding. The PNA's unfavorable pharmacologic and pharmacokinetic properties for antisense approaches were overcome by using PNA as a complex with ASO.

In fact, our gene ontology (GO) term analysis of the proteins that bind to LNA-gapmers revealed that many were RNA-binding proteins (RBPs) (Fig. 5b). For example, Hnrnpk[32], Hnrnpa2b1[57], Pcbp2[31], Fus[33], and Hmgb1[58], which are identified as ssASO-binding proteins (Supplementary Table 2), are well described to recognize the nucleobase, ribose, and phosphate backbone moieties. Thus, BS is expected to effectively disrupt the hydrogen bonding and electrostatic and hydrophobic interactions formed between ssASOs and these proteins. The difference in the nuclear distribution pattern of ssASO and its BRO also supports that ssASO is inadvertently incorporated into organelles by interacting with RNAs or proteins, suggesting that BS helps alter their localization pattern. Indeed, the moderately toxic mPCS1 appeared to localize mainly to the nucleolus, whereas the corresponding non-toxic BRO was found to be distributed within the nucleoplasm, avoiding the nucleolus.

The mPCS1f ssASO and the mPCS1f/PNA(C8) BRO used in the cellular distribution study in Fig. 5e display a sequence divergence of more than three bases from human *PCSK9* mRNA, and this homology constraint precludes us from evaluating their knockdown efficacy within this human cell line context; nevertheless, the activity of the BRO was comparable to that of the ssASO in mice (Fig. 1c). At the same time, while the mPCS1 ssASO manifests discernible toxicity in both mice and human cell lines, the BRO largely obviates such toxicities (Fig. 1d and Fig. 5b, c). Therefore, citing a work by Liang et al. [59], which has shown the possibility of ssASOs operating in both the cytoplasm and the nucleus, we postulate that the observed intracellular distribution variance between mPCS1 ssASO and its BRO counterpart may predominantly influence toxicity rather than therapeutic activity.

This narrative aligns with the widely recognized mechanism proposed by Crook et al. [60,61], which implicates the ssASO-protein interactions lead to disruption of cellular homeostasis. In this context, a possibility for nucleolar transport of ssASO may be due to the previously-observed interaction of ssASO with paraspeckle proteins such as P54nrb (also known as Nono), PSF, and FUS[13]. However, this study did not observe the apparent co-localization of P54nrb and mPCS1f ssASO in the nuclei (Supplementary Fig. 5). Of note, this study differs from previous studies as ASOs were introduced into cells using the CEM method, rather than conventional lipofection. These observations raise the question of whether any other specific factors, such as RNAs and proteins listed in Supplementary Data 1, dictates these intracellular distribution shifts. We aspire to delve deeper into the fundamental mechanisms responsible for toxicity, employing BRO as a tool.

Another observation pertains to the protein binding pattern when parallel-type PNA is applied to the 18-mer long ASO, as shown in Fig. 7f. In this gel pattern, there appears to be a slight increase in binding to low-molecular-weight proteins as the PNA lengthens, although not to the same extent as with ssASO. The rigid and uniform double-stranded

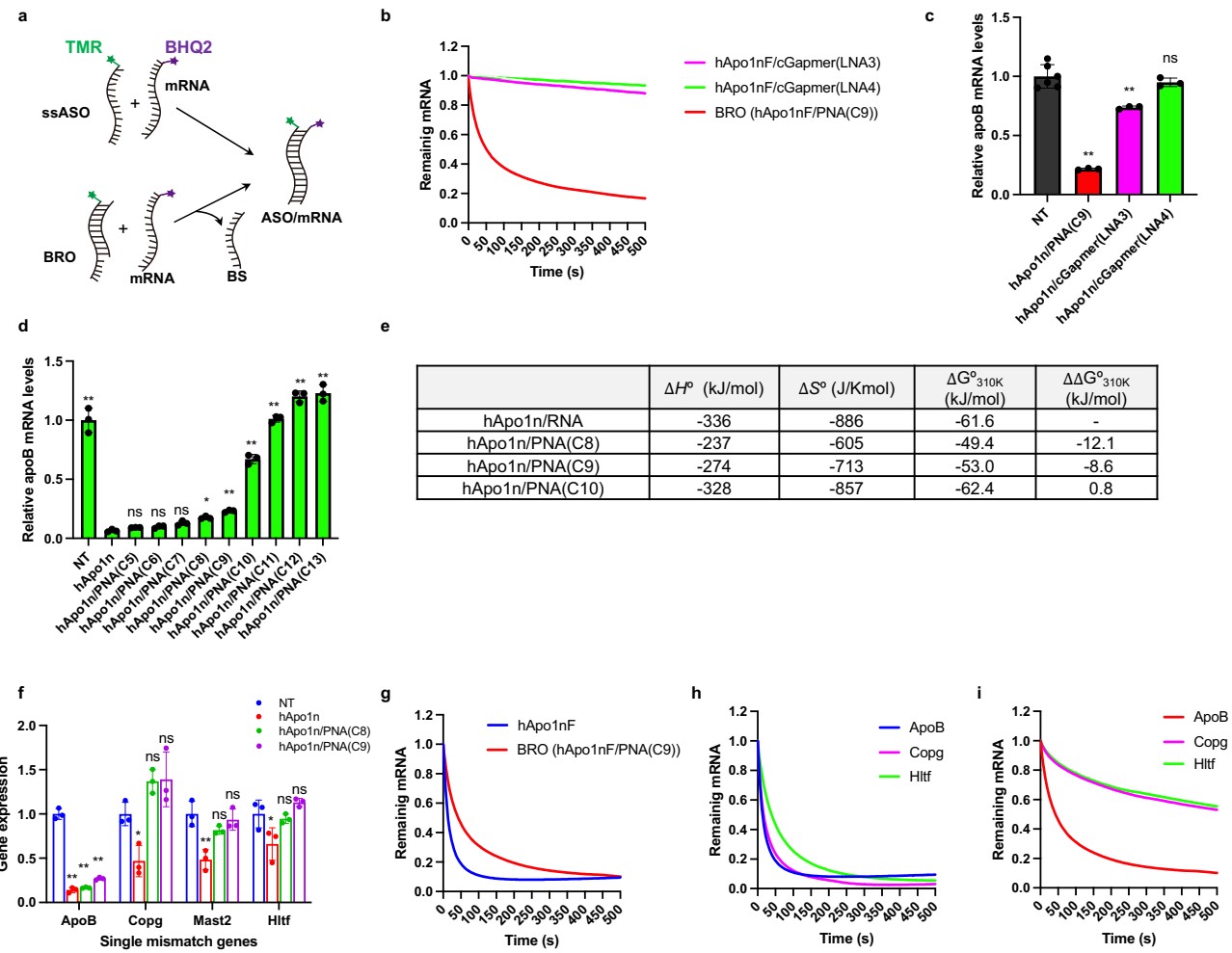

**Fig. 6 | Complementary PNAs suppress off-target hybridizations. a** Schematic illustration of an in-cuvette FRET assay. **b** Hybridization kinetics were evaluated for the indicated ssASO and BROs using FRET ($n = 3$ independent experiments). **c** The antisense effects of hApo1n/PNA(C9) and hApo1n/cGapmer on *ApoB* mRNA in Huh-7 cells using the CEM transfection method ($n = 6$ biologically independent samples for control, $n = 3$ for BRO series; one-way ANOVA followed by two-sided Dunnett's multiple comparison tests comparing to control, $p$ values from left side; <0.0001, 0.0006, 0.6547). **d** The antisense effects of hApo1n and hApo1n/PNA(C5-C13) BROs in Huh-7 cells using the CEM transfection method ($n = 3$ biologically independent samples; one-way ANOVA followed by two-sided Dunnett's multiple comparison tests comparing to hApo1n, $p$ values from left side; <0.0001, 0.9854, 0.99398, 0.4175, 0.0388, 0.0011, <0.0001, <0.0001, <0.0001, <0.0001). **e** The van't Hoff analysis for the indicated ssASO and BROs ($n = 3$ independent experiments).

**f** Knockdown activities of on-target *ApoB* gene and three single-mismatch genes were compared for indicated hApo1n and BROs (hApo1n/PNA(C8), hApo1n/PNA(C9)) in Huh-7 cells exposed to 1 μM after 24 h of CEM transfection ($n = 3$ biologically independent samples; one-way ANOVA followed by two-sided Dunnett's multiple comparison tests comparing to control, $p$ values from left side; ApoB: <0.0001, <0.0001, <0.0001; Copg: 0.0299, 0.1307, 0.1071; Mast2: 0.0012, 0.1826, 0.8253; Hltf: 0.0262, 0.9132, 0.4780). **g** Hybridization kinetics for on-target mRNA (*ApoB*) were evaluated for the indicated ssASO and BROs using FRET ($n = 3$ independent experiments). **h** FRET assay of TAMRA-labeled hApo1nF and BHQ2-labeled mRNA mimics (Copg, Hltf). **i** FRET assay of TAMRA-labeled hApo1nF/PNA(C9) BRO and BHQ2-labeled mRNA mimics (Copg, Hltf). Data in Fig. 6b–d and g–i are presented as mean values ± SD. **$p < 0.01$, *$p < 0.05$. "ns" indicates not significant ($p > 0.05$). Source data are provided as a Source Data file.

structure of the long ASO, in conjunction with hydrophobic PNA, may potentially promote an increase in protein binding capacity. Furthermore, as depicted in the 3D model in Fig. 7a (see also Supplementary Fig. 18 and Supplementary Data 3), BRO employing parallel-type PNA possesses a structure where the major groove is widely opened, which might serve as an interaction site with proteins. Despite these observations, toxicity was not detected in this instance (Fig. 7c), suggesting that this protein interaction does not compromise safety, either in quality or quantity. Ongoing analysis is underway to identify what proteins and how they might bind to BRO.

We identified several other RBPs associated with RNA processing, splicing, and transcription as ssASO-binding proteins (Fig. 5d). In addition to RBPs, enzymes involved in apoptosis and innate immunity have been identified as ssASO-binding proteins (Supplementary Data 1). Disturbance of these enzymes by ASOs can affect the global mRNA expression profile, possibly triggering massive cellular stress.

In fact, our GSEA analysis of the RNA-seq data suggested that ssASO-induced oxidative stress, proteasomal degradation of proteins, immune responses, inhibition of mitochondrial functions, and cell death responses. An apoptosis pathway was enriched, suggesting a possible form of cell death caused by ASOs as suggested by earlier studies. Reverting the cellular metabolism from glycolysis-dominated to oxidative phosphorylation (OXPHOS) in conventional cultured cells with a galactose-supplemented medium in replacement of glucose enables the restoration of cellular sensitivity to drugs[62]. Such in vitro cell culture conditions allowed us to uncover the apoptosis-inducing ability of ASOs, where ASOs exhibited the apoptosis-related mitochondrial dysfunction/caspase activation capabilities[63].

Further investigations are required to elucidate the mechanisms underlying the low-intensity toxicity and to identify specific markers for its monitoring and prediction. As evidenced in the mPCS1 study, no direct apoptotic phenotype was associated with low-intensity ALT

**Table 3 | Sequence used in this study and melting temperature ($T_m$) of ASO/RNA or ASO/PNA**

| No. | Position ID | Sequence | −3 | −2 | −1 | 1 | 2 | 3 | 4 | 5 | 6 | 7 | 8 | 9 | 10 | 11 | 12 | 13 | 14 | 15 | 16 | 17 | 18 | | $T_m$ (°C)[a] | Target |
|---|---|---|---|---|---|---|---|---|---|---|---|---|---|---|---|---|---|---|---|---|---|---|---|---|---|---|
| 1 | Acsl1 | 5' | Y | Y | Y | C | T | C | c | a | t | g | a | c | a | c | a | g | c | a | T | T | a | 3' | | Acsl1 (human/mouse) |
| 2 | Acsl1n | | | | | C | T | C | c | a | t | g | a | c | a | c | a | g | c | a | T | T | a | | | |
| 3 | Acsl1b | | | | b | C | T | C | c | a | t | g | a | c | a | c | a | g | c | a | T | T | a | | | |
| 4 | pPNA(C14)-Acsl1 | N-term | | | | | | | | pT | pA | pC | pT | pG | pT | pG | pT | pC | pG | pT | pA | pA | pT | C-term | 52 ± 0.3 | |
| 5 | pPNA(C15)-Acsl1 | | | | | | | | pG | pT | pA | pC | pT | pG | pT | pG | pT | pC | pG | pT | pA | pA | pT | | 55 ± 0.4 | |
| 6 | pPNA(C16)-Acsl1 | | | | | | | pG | pG | pT | pA | pC | pT | pG | pT | pG | pT | pC | pG | pT | pA | pA | pT | | 61 ± 0.8 | |
| 7 | pPNA(C17)-Acsl1 | | | | | | pA | pG | pG | pT | pA | pC | pT | pG | pT | pG | pT | pC | pG | pT | pA | pA | pT | | 67 ± 0.9 | |
| 8 | pPNA(C18)-Acsl1 | | | | | pG | pA | pG | pG | pT | pA | pC | pT | pG | pT | pG | pT | pC | pG | pT | pA | pA | pT | | 73 ± 0.2 | |
| 9 | cRNA(18)-Acsl1 | 3' | | | | rG | rA | rG | rG | rU | rA | rC | rU | rG | rU | rG | rU | rC | rG | rU | rA | rA | rU | 5' | 65 ± 0.2 | |

N: LNA, n: DNA, Y: GalNAc$_{APD}$, pN: PNA, rN: RNA, b: biotin.
[a]$T_m$ vs Acsl1n (4 µM).

elevations (Supplementary Figs. 2–4). Shen et al. also indicated that ASOs with ALT levels between 600 and 7000 U/L do not consistently exhibit apoptosis phenotypes[13]. Thus, it remains ambiguous whether the primary cause of ASO hepatotoxicity can be attributed exclusively to apoptosis. In the current investigation, the marked upregulation of mRNA at the *Mlkl* level, combined with activation of transcription factors associated with inflammation, was observed even at modestly elevated ALT levels (~300 U/L), suggesting a possible role of necroptosis. Recently, necroptosis has been highlighted as another major contributor to programmed cell death in the liver and other tissues[64]. In necroptosis, the pseudokinase mixed lineage kinase domain-like proteins (MLKLs) associate with a strong pro-inflammatory signal, leading to tissue damage with massive inflammation[65]. Such pathways associated with low-intensity toxicity, or changes that precede apoptosis as identified by the BRO strategy, should provide deeper insights into the underlying upstream triggers of toxicity in the near future.

### Hybridization-dependent mechanism

Another important aspect of this study is how BS affects the target hybridization of ASO. To verify whether BRO-formulated ASO hybridization is catalyzed by its toehold, we employed artificial nucleic acids to generate toehold-lacking incumbent strands with the same binding affinity with toehold-bearing PNA. To the best of our knowledge, this study is a rare and compelling example of resolving the thermodynamic and kinetic entanglements often associated with toehold chemistry and successfully distilling the kinetic effects of toehold through the use of artificial nucleic acids[66]. Furthermore, our data supports that this mechanism operates to elicit antisense activity within biological systems. One of the requirements that the BRO system must meet to facilitate the TMSD reaction is that the binding free energy of ASO for PNA must be smaller than that for RNA, which does not violate the fundamental laws of thermodynamics.

On the other hand, although the Gibbs free energy ($\Delta G_{310K}$) values of hApo1n/PNA(C8) and hApo1n/PNA(C9) are both smaller than those of the hApo1n/target RNA duplex (Fig. 6e), the hApo1/PNA(C8) BRO duplex had a higher ALT value. This result suggests that the hApo1/PNA(C8) combination has basal leakage of toxic ssASO, which might be due to insufficient hybridization under physiological conditions. Thus, the thermodynamic stability of the BRO duplex should meet the minimum requirement of free energy for stability under biological conditions (Supplementary Fig. 6). From the results shown in Fig. 6e, adjusting the stability of ASO/BS to approximately −8 to −10 kJ/mol from the $\Delta G_{310K}$ value between ASO/mRNA can possibly promote the

active involvement of TMSD. We also should care that an exaggerated affinity of ASO against target RNA has been demonstrated to be detrimental to off-target toxicity[27,67,68].

The optimal balance between the affinity of ASO for RNA, for PNA, and toehold domain lengths must be sought to ensure that this system is properly operated in vivo. To best adjust the affinity and coverage, two approaches can be used: (1) enthalpy effects (e.g., the use of non-canonical electrostatic and hydrogen bonding interactions) and (2) entropy effects (e.g., the use of the chiral amplification effect[69] by incorporating chiral PNA analogs and/or other chiral sources in the achiral strand). The parallel binding mode of PNA was also demonstrated to be effective for longer ASOs. Especially in the case of parallel BRO strategy, off-target hybridization of BS to potentially existing antiparallel RNAs is a secondary event that should be taken into an account. However, no resulting toxicity phenotype was observed in this study due probably to the originally unfavorable pharmacological and pharmacokinetic properties of PNA as an ASO.

It is also intriguing that the boost in activation energy due to the negative catalytic effect of BS on the ASO hybridization reaction can retard off-target hybridization. Our results have demonstrated the utilization of TMSD can limit ASO binding to at least certain single-mismatch-containing RNAs, indicating that hybridization-dependent off-target interactions can be suppressed through this physicochemical mechanism. A more expansive understanding of the effect of the BRO's mismatch recognition ability on the suppression of hybridization-dependent knockdown, along with a more intricate exploration of the impact of hybridization-dependent knockdown on toxicity, remains an area in need of further investigation[24]. Of note, the mismatch discrimination ability of the TMSD reaction is known to depend on the location of the mismatch[41]. Thus, whether BRO follows this rule in vivo will be an interesting finding.

One caveat of our present investigation is its predominant emphasis on LNA gapmers, thereby leaving room for the exploration of the applicability of the BRO strategy for ASOs with other RNA chemistries, such as MOE or 2'OMe, which have earned therapeutic approvals. Nevertheless, as a first approximation, the physicochemical principles presented herein are expected to possess considerable universality. This infers that the BRO architecture could feasibly be adapted to gapmers or splice-switching ASOs based on RNA analogs distinct from LNA, with the potential to confer comparable benefits. However, since the duplex-stabilizing ability of these RNA analogs is smaller compared to that of LNA, distinct optimal BS attributes (length/position/orientation) might be exhibited from those of LNA

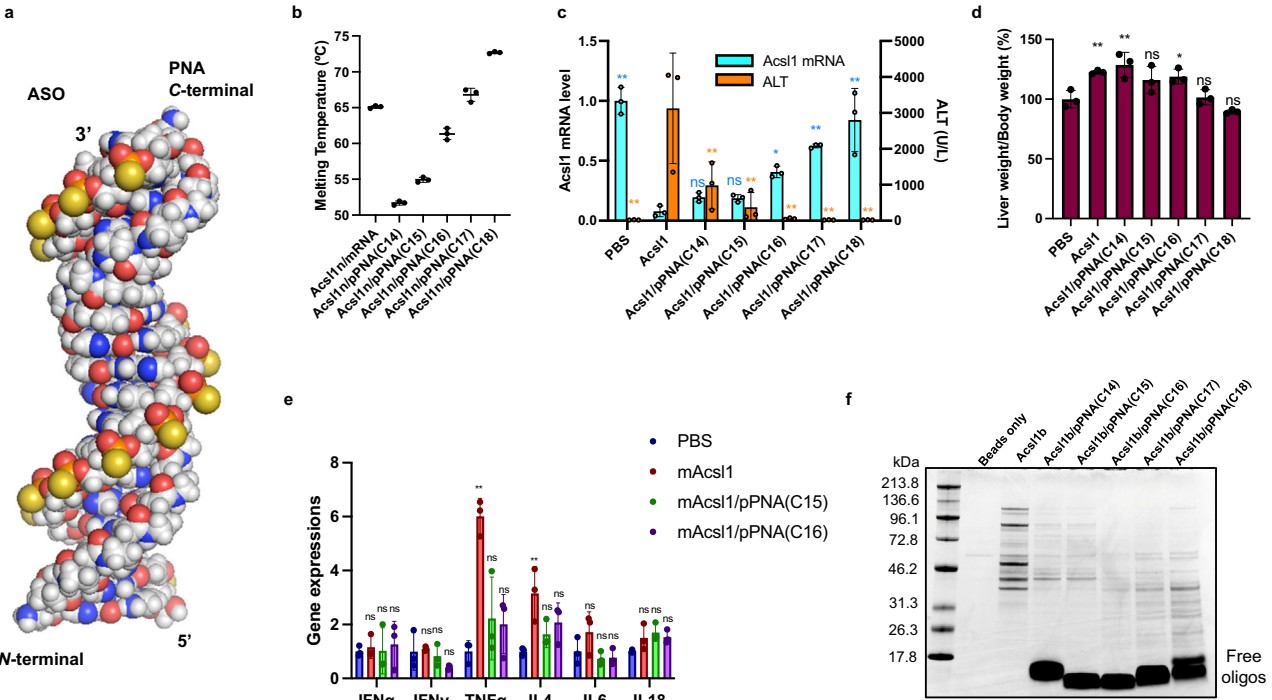

**Fig. 7 | Parallel PNAs as brother strand (BS) have identical effect on ssASO.**
Molecular modeling of Acsl1/parallel PNA **a** duplex constructed by optimization
using the GFN1-XTB method (Supplementary Fig. 18 and Supplementary Data 3).
**b**–**e** Mouse was subjected to a single subcutaneous injection of indicated ssASO
and BROs at a dose of 150 nmol/kg. Melting temperatures of duplexes indicated
($n = 3$ independent experiments) (**b**). *Acsl1* mRNA expression and ALT levels ($n = 3$
biologically independent samples; one-way ANOVA followed by two-sided Dun-
nett's multiple comparison tests comparing to Acsl1, *p* values from left side; Acsl1
mRNA: <0.0001, 0.6407, 0.6986, 0.0144, 0.0002, <0.0001; ALT: 0.0002, 0.0059,
0.0007, 0.0002, 0.0002, 0.0002) (**c**), liver weight ($n = 3$ biologically independent

samples; one-way ANOVA followed by two-sided Dunnett's multiple comparison
tests comparing to control, *p* values from left side; 0.0096, 0.0014, 0.0760,
0.0308, 0.9996, 0.3554) (**d**), and liver cytokine mRNA levels ($n = 3$ biologically
independent samples; two-way ANOVA followed by two-sided Dunnett's multiple
comparison tests comparing to control, *p* values from left side; <0.0001, 0.0572,
0.1375, 0.0004, 0.4732, 0.1077, 0.3809, 0.9160, 0.9453, 0.6497, 0.3958, 0.6092) (**e**)
were evaluated 96 h after administration. **f** Identification of liver-derived proteins
bound by CBB staining for Acsl1b or Acsl1b/pPNA(C14–18) BROs. Data in Fig. 7b–e
are presented as mean values ± SD. \*\**p* < 0.01, \**p* < 0.05. "ns" indicates not sig-
nificant (*p* > 0.05). Source data are provided as a Source Data file.

gapmers. Further detailed exploration tailored to diverse chemistries
and modalities is anticipated in future research.

A lingering question in this study pertains to the impact of the
PNA brace on the in vivo pharmacokinetics and cellular uptake
mechanisms of ligand-free ASOs. Given the recent emphasis on ASO
ligand conjugation technologies, our primary focus in the in vivo
research centered on ligand-conjugated ASOs[5]. This approach helped
mitigate the challenges of directly comparing ssASOs with BROs, as
ligand-conjugated ASOs tend to be less influenced by the differential
dynamics between these two entities. Yet, as highlighted by Liang
et al.[70], the intracellular activity of ASOs is profoundly influenced by
proteins binding to PS-ASO. We hypothesize that the minimization of
hybridization-independent interactions in BROs could alter their
in vivo pharmacokinetics as well as the underlying cellular uptake
mechanisms. This intriguing angle warrants exploration in future
research.

In this study, we tried to tackle the long-standing safety challenge
of ASO drugs by leveraging thermodynamics and DNA nanotechnol-
ogy insights. The present study revealed that our BROTHERS tech-
nology enables the extension of the therapeutic window of
conventional gapmer ASOs. The ability of BRO to suppress both
inseparable off-target interactions supposedly contributed to its
safety. This technique can not only help uncover the toxicity
mechanism of ASOs, but also produce a BRO class of drugs as well as
revive ASOs whose development has been halted due to safety con-
cerns. Ensuring the safety margin would also have a transformative
impact on the extrahepatic delivery of ASOs as organs and tissues
besides the liver might be less active for the uptake of molecules.

## Methods

### Ethical statement
All animal experiments were performed in accordance with the
guidelines for animal experimentation of Nagasaki University (Naga-
saki, Japan), with the consent of the Animal Care Ethics Committee
(approval number:1911011572-6).

### Reagents
Sequences and chemistries of all tested ASOs and PNAs are listed in
Supplementary Table 4. Information about antibodies and primer-
probe sets for qRT-PCR is listed in Supplementary Table 5, 6. Huh-7
cells (JCRB number: JCRB0403) were purchased from JCRB Cell Bank
(Japanese Collection of Research Bioresources Cell Bank).

### Solid phase synthesis of antisense oligonucleotides and PNA oligomers
Oligonucleotides listed were synthesized using an automated DNA/
RNA synthesizer, and the remainder was outsourced to GeneDesign,
Inc. (Osaka, Japan). The oligonucleotides were synthesized at a
1.0 μmol scale using standard solid-phase oligonucleotide synthesis
with 5-benzylthio-1*H*-tetrazole (0.25 M in MeCN) as an activator.
DMTr-protected ASOs were treated with 28% aqueous NH₃ at 55 °C for
12–14 h to cleave oligomers from the CPG, and all the protecting
groups except the terminal DMTr group were removed. For the
deprotection of the 5′-DMTr group, ASOs were treated with 4% (v/v)
aqueous trifluoroacetic acid on a Glen-Pak™ DNA purification cartridge
(Sterling, VA). The resulting ASOs were further purified using reverse-
phase HPLC. The purity of the materials was analyzed using analytical

reverse-phase HPLC, and identification was performed using MALDI-TOF-MS (Supplementary Fig. 10). PNA oligomers were synthesized using a home-built parallel PNA synthesizer, following Fmoc-based PNA synthesis on a 1 or 5 µmol scale (Supplementary Table 2, Supplementary Figs. 7–9). PNA was deprotected and cleaved from the solid support through TFA-TIS-H$_2$O (95/2.5/2.5, v/v) treatment for 90 min at 25 °C. The cleavage solution was collected in a tube and the PNA product was precipitated in ice-cold diethyl ether and washed thrice through centrifugation. Crude PNAs were purified using reverse-phase HPLC. The purity of the materials was analyzed using analytical reverse-phase HPLC, and identification was performed using MALDI-TOF-MS. Cy5-labeled PNA(C9)F was obtained from PANAGENE (Daejeon, South Korea). (See Supplementary Information for more detail).

## Animal studies

C57BL/6 J mice were purchased from SLC Japan (Tokyo, Japan). All mice were male, and all experiments commenced when the animals were 6–8 weeks old. The mice were housed in standard rodent cages in a room kept at 25 °C and 60% humidity, under a 12-h light/dark cycle. The mice had ad libitum access to water and feed (CE-2, CLEA Japan, Tokyo, Japan) for 1–2 weeks before the experiments at the animal facility of Nagasaki University. Equimolar mixtures of ASO and PNA were annealed through bathing overnight in hot water and then cooling slowly. ASOs or PBS were subcutaneously administered to the mice. Seventy-two hours after a single administration, the mice were anesthetized with isoflurane (Escain®, Pfizer Japan, Tokyo, Japan) and sacrificed to collect the livers and kidneys. Peripheral blood was collected in BD Microtainer ® tubes (BD, Franklin Lakes, NJ) to separate serum. Serum AST/ALT was measured using a Fuji dry-chem slide GPT/ALT-PIII or outsourced to Oriental Yeast Co., Ltd. (Tokyo, Japan). Total cholesterol levels were measured using TCHO-PIII (Wako, Japan), and an aliquot of 10 µL of each serum sample was used. Total RNA was extracted using RT-S2 (Wako, Japan), according to the manufacturer's protocols. RNA samples were reverse-transcribed to obtain cDNA using a High-Capacity cDNA Reverse Transcription Kit (Applied Biosystems) and quantitative PCR was performed using Fast SYBR® Green Master Mix (Thermo Fisher Scientific). *ApoB* mRNA levels were normalized to GAPDH mRNA levels. Information regarding the primer-probe sets for qRT-PCR is listed in Supplementary Information. The ED50 values were estimated with nonlinear regression (three parameters) using GraphPad Prism9 software (Version 9.4.1).

## RNA-seq analyses

Total RNA was quantified using the Qubit™ RNA High Sensitivity (HS) and Broad Range (BR) Assay Kits (Thermo Fisher), further diluted to 2.5 ng/µL, and analyzed using a Bioanalyzer RNA 6000 pico assay to calculate the RNA integrity number (RIN) (See Supplementary Information) (Supplementary Fig. 11). RNA-seq samples were prepared using total RNA with high RIN from mouse livers with Collibri™ 3′ mRNA Library Prep Kit for Illumina™ Systems (Life Technologies, # A38110024), following the manufacturer's instructions. Sequencing data were obtained using Novaseq 6000 (6 gb/sample) (Supplementary Figs. 12, 13). The RNA-seq data generated in this study have been deposited in the DDBJ Sequence Read Archive under the bioproject accession code "PRJDB15572". Created Fastq files were processed on the RaNA-seq website (https://ranaseq.eu), and DE and functional enrichment analyses were performed with DESeq2 using the Wald test (cutoff = 0.05)[71]. wPGSA was performed using RNA-seq fold change (FC) data at http://wpgsa.org[47]. The relative Enrichment Score was calculated by dividing the enrichment score of ssASO by that of BRO.

## In silico sequence analysis

In silico sequence analysis was performed using GGGenome (https://GGGenome.dbcls.jp/), which allows a comprehensive search for mismatches, insertions, or deletions. The database we used here is mouse pre-spliced RNA, with RefSeq curated protein coding on GRCm39/mm39 and D3G 22.02 (Feb, 2022). The measure we used to define the total number of mismatches, insertions, and deletions between ASO and the complementary RNA sequences was "distance (*d*)", as introduced by Yoshida et al. [46]. (Supplementary Table 3).

## In vivo imaging

All mice used in this assay were 6-week-old male Balb/cSlc-nu/nu mice purchased from SLC Japan (Tokyo, Japan). Mice were fed an autofluorescence-reduced diet (D10001, Research Diets Inc.) for more than 1 week before the assays were conducted. Cy5-labeled oligonucleotides (BRO (hApo1/PNA(C9)F) and PNA(C9)F) were dissolved in saline for injection. For in vivo imaging, mice were intravenously injected with a single dose of each oligonucleotide (300 pmol in 200 µL saline) through the tail vein, and the biodistribution was visualized using an IVIS Lumina II imaging system (Caliper Life Science; excitation filter, 640 nm; emission filter, Cy5.5., exposure time = 5 s) at different time points for up to 1 h (n = 3). For ex vivo imaging, the mice were dissected after 5 min (n = 1). Fluorescence images of the ex vivo organs were visualized using an IVIS Lumina II imaging system (excitation filter, 640 nm; emission filter, Cy5.5. and an exposure time of 1 s). The mice were anesthetized with isoflurane to capture snapshots.

## Western blotting

Distinct Liver samples (n = 3) were homogenized using bead crushers µT−12 (TAITEC) in RIPA buffer supplemented with a 1× protease inhibitor cocktail (both from Nacalai Tesque). Total proteins were quantitated using a bicinchoninic acid protein assay (Takara Bio Inc.), and 50 µg total protein was added to 1 M DTT and heated to 95 °C for 3 min for thermal denaturation. Proteins were separated on a 4–20% gradient SDS-PAGE gel and transferred to a nitrocellulose membrane using an iBlot Gel Transfer Device (Thermo Fisher Scientific). Membranes were blocked with Blocking One (Nacalai Tesque) at room temperature for 30 min and incubated with primary antibodies in blocking buffer at room temperature for 1 h or at 4 °C overnight. After washing thrice for 10 min with PBS-T, the membranes were incubated with secondary antibodies in blocking buffer at room temperature for 1 h. Then, membranes were washed thrice with PBS-T, and proteins were detected using enhanced chemiluminescence (Thermo Fisher Scientific). Images were collected using the Bio-Rad Molecular Imager ChemiDoc Touch (Supplementary Fig. 14).

## Cell culture and treatment

Huh-7 cells, a human hepatoma cell line, was cultured at 37 °C, 5% CO$_2$ in Dulbecco's modified Eagle's medium (DMEM; Sigma-Aldrich) supplemented with 10% heat-inactivated fetal bovine serum (FBS) and 1% penicillin/streptomycin. Huh-7 cells were seeded at 10,000 cells/well in 9 mM Ca$^{2+}$ enrichment medium (CEM) containing ASOs or BRO at a final concentration of 1 µM in 96-well plates. After 24 h, cells were washed with PBS, and cDNA was synthesized directly from cell lysates using a SuperPrep Cell Lysis & RT Kit for qPCR (Toyobo). qRT-PCR was performed using Fast SYBR® Green Master Mix (Thermo Fisher Scientific) and analyzed using the StepOnePlus system (Applied Biosystems).

## Caspase 3/7, 9, and 8 assays

For caspase 8, 9, and 3/7 assays, 5000 cells were transfected with 2 µM mPCS1n or BRO (mPCS1n/PNA(C8)) using the CEM method with 10 mM galactose medium substituted for glucose medium. After 48 h of incubation at 37 °C with 5% CO$_2$, cells were washed with PBS, and medium (100 µL) was replenished. Caspase-Glo reagent (100 µL; Promega) was added directly to the cells in a 96-well plate. After 30 min of incubation at room temperature, the luminescence was recorded using a multimode microplate reader (Agilent Technologies). Background readings were measured from wells containing CEM medium

without cells and subtracted from the control or assay values. Relative caspase activity was calculated as the luminescence reading of a treated sample/luminescence reading of a non-treated control.

## Cell proliferation assay

For the cell viability assay, 5000 cells were transfected with 0.125–2 μM mPCS1n or BRO (mPCS1n/PNA(C8)) using the CEM method with glucose medium or 10 mM galactose medium substituted for glucose medium. After 72 h, the cells were washed with 1× PBS, and the medium was replenished. Cell counting kit-8 reagent (10 μL; Dojindo) was added directly to a 96-well plate. After 2 h incubation at 37 °C and 5% $CO_2$, 450 nm absorbance was measured using a multimode microplate reader. Background readings were measured from wells containing CEM medium without cells and subtracted from the control or assay values.

## PI staining

For the dead cell staining assay, 5000 cells were transfected with 2 μM mPCS1n or BRO (mPCS1n/PNA(C8)) using the CEM method with glucose medium or 10 mM galactose medium substituted for glucose medium. PI (Dojindo) was added directly to the cells at a final concentration of 1 μM. After incubation for 15 min at 37 °C and 5% $CO_2$, PI fluorescence ($\lambda_{ex}$ = 530 nm, $\lambda_{em}$ = 620 nm) was visualized using a fluorescence microscope (Carl Zeiss Co., Ltd.).

## JC-1 MitoMP detection assay

Cells (5000) were transfected with 2 μM mPCS1n or BRO (mPCS1n/PNA(C8)) using the CEM method with glucose medium or 10 mM galactose medium substituted for glucose medium. After 48 h, the JC-1 reagent (Dojindo) was added to the wells at a final concentration of 2 μM. After incubation for 2 h at 37 °C and 5% $CO_2$, cells were washed with 1× HBSS and the imaging buffer solution was added. Cells were observed using a fluorescence microscope (Axio Vert.A1, Carl Zeiss Co., Ltd.). Fluorescence imaging: green λex = 488 nm, λem = 500–550 nm, red $\lambda_{ex}$ = 561 nm, λem = 560–610 nm. Fluorescence images were analyzed by ZEN software 3.3. 89. 0000 (blue edition).

## Immunofluorescence

Cells seeded on collagen coating glass-bottom culture dishes (Matsunami Glass Ind., Ltd.) were transfected with 250 nM mPCS1f or BRO (mPCS1f/PNA(C8)) and incubated for 48 h at 37 °C and 5% $CO_2$. After rinsing thrice with 1× PBS, cells were fixed at room temperature with 4% formaldehyde for 10 min and permeabilized with 0.1% NP-40 in 1× PBS for 10 min. Fixed cells were blocked in Blocking One Histo (Nacalai Tesque) at room temperature for 30 min and then incubated in 20× diluted Blocking One Histo (Nacalai Tesque) with primary antibodies at room temperature for 2 h. After washing thrice with PBS-T (5 min per wash), cells were incubated in 20× diluted Blocking One Histo with secondary antibodies at room temperature for 1 h. Finally, cells were washed thrice with PBS-T (5 min per wash) and nuclei were stained with Hoechst 33258 (Dojindo) for 10 min. The cells were mounted using Fluoromount (Diagnostic Biosystems). Confocal images were generated using confocal laser scanning microscopy (LSM710 microscope, Carl Zeiss Co., Ltd.) and analyzed by ZEN software 3.5. 093. 00001 (blue edition). The 3D structure was rendered in PyMol (Version 2.4.1).

## Protein pull-down assay and in-gel digestion

Streptavidin magnetic beads (0.2 mg per reaction; TAMAGAWA SEIKI Co., Ltd.) were washed thrice with 1× PBS and incubated with 100 μM 5′-biotinylated to capture ASOs in 1× PBS (200 μL) at 4 °C for 1 h. The beads were washed thrice with binding buffer (20 mM HEPES-NaOH (pH 7.9), 100 mM KCl, 1 mM $MgCl_2$, 0.2 mM $CaCl_2$, 0.2 mM EDTA, 10% (v/v) glycerol, 0.1% (w/v) NP-40, 1 mM DTT, and 0.2 mM PMSF) and aliquoted. After adding 2 nmol of complementary strand PNA to the

beads, the beads were incubated at 4 °C for 2 h with 450 μg of mouse liver lysate in binding buffer. For the mPCS1 series, the beads were washed thrice with binding buffer and incubated with 0.5 nmol competitor non-biotinylated ASOs or BRO for 5 min at room temperature. Supernatants were collected as the eluted proteins and beads were incubated with 1.5 nmol competitor non-biotinylated ASOs or BRO again. For the hApo1 and Acsl1 series, the beads were washed thrice with binding buffer and incubated with 1.0 nmol competitor non-biotinylated ASOs or BRO for 5 min at room temperature. Eluted proteins were separated on 4–12% or 4–20% gradient SDS-PAGE gels. CBB staining was performed to visualize the binding proteins (Supplementary Figs. 15–17). Protein pull-down assay was performed repeatedly ($n$ = 2 or $n$ = 3). The 30 kDa-70 kDa portion of the gel for mPCS1 or BRO was cut out where there was a large difference in the band intensity. Gels were decolorized with 25 mM $NH_4HCO_3$ and 50% MeCN (200 μL) for 10 min, repeated three times at room temperature. The gels were then dehydrated in MeCN (100 μL) for 10 min and subsequently vacuum dried for 10 min. After 25 mM $NH_4HCO_3$ solution (100 μL) containing 10 mM DTT was added to the gels, the gels were further incubated at 56 °C for 45 min. After removing the solution, the gels were washed with 25 mM $NH_4HCO_3$ solution for 10 min. For alkylation, gels were incubated in 55 mM iodoacetamide and 25 mM $NH_4HCO_3$ solution (100 μL) for 45 min at room temperature, protected from light. This was followed by another wash in 25 mM $NH_4HCO_3$ (100 μL) for 10 min at room temperature. Gels were again dehydrated using 25 mM $NH_4HCO_3$ and 50% MeCN (200 μL), repeated three times, and then once more with MeCN (100 μL) for 10 min. After vacuum drying for 10 min, proteins in the gels were digested with trypsin (Wako, 10 μg/mL in 50 mM $NH_4HCO_3$) for 30 min on ice, followed by 12 h at 37 °C. The resulting peptides were eluted from the gels using 0.1% TFA and 50% MeCN, repeated three times for 30 min each. The eluted peptide solution was concentrated to less than 10 μL and desalted using ZipTip C18 (Millipore), and the extracted peptide fragments were analyzed using nano-LC-MS/MS. The peptide mixture (2 μL) was injected into the injection loop of a nano-precolumn (Acclaim PepMap™ 100, 75 μm x 2 cm, nano Viper, C18, 3 μm, 100 Å, Thermo Fisher Scientific) and washed with 0.1% trifluoroacetic acid in 2% acetonitrile. An MS/MS instrument (Q-Exactive series, Thermo Fisher Scientific) equipped with a nano-LC system (EASY-nLC™, Thermo Fisher Scientific) was used for analysis. Peptides were separated on a nano-LC column (C18, 75 μm i.d. × 125 mm, 3 μm particle, 100 Å pore size, Nikkyo Technos) with gradient elution and ion-sprayed into the MS/MS instrument using a spray voltage of 2.0 kV. All spectra were measured with an overall mass/charge ratio range of 400–1500. Full MS scans were acquired with a mass resolution of 70,000. Tandem mass spectra were acquired with a mass resolution of 17,500. Spectra were searched against sub-databases from the public nonredundant protein database UniProt Knowledgebase (mouse) with the following search parameters: mass type, monoisotopic precursor and fragments; enzyme, trypsin (KR); enzyme limits, full enzymatic cleavage allowing up to two trypsin missed cleavages sites; peptide tolerance, 10 ppm; fragment ion tolerance, 0.8 Da; ion and ion series calculated, B and Y ions; static modification, C (carbamidomethylation, +57.021 Da); and differential modifications, M (oxidation, +15.995 Da), N, and Q (deamidation, +0.984 Da). MS/MS data were extracted using Proteome Discoverer ver. 1.4.0.288 (Thermo Fisher Scientific).

## Melting temperature measurement and thermodynamic analysis

UV melting experiments were performed using a Shimadzu UV-1850 spectrophotometer equipped with a TMSPC-8 $T_m$ analysis system (Shimadzu). Two complementary single-stranded oligonucleotides or PNA were mixed in a 10 mM sodium phosphate buffer (pH 7.0) containing 100 mM NaCl and 0.1 mM EDTA to obtain a 2 or 4 μM oligonucleotide solution. The mixture was annealed by heating at 95 °C for

3 min, and then cooled at 1 °C /min to 20 °C. $T_m$ was estimated using the software provided with TMSPC-8 and defined as the temperature at which the formed duplexes were half-dissociated. The temperature-dependent change in the optical absorption was recorded at 260 nm, and the scan rate was 0.5 °C/min, ranging 4–94 °C. Each averaged $T_m$ value was determined as the intersection point between the absorption of temperature and the median between the two baselines from three independent measurements.

The thermodynamic parameters ($\Delta H°$ and $\Delta S°$) for the formation of RNA/DNA hybrids or PNA/DNA hybrids were determined from the $T_m^{-1}$ versus $\ln(C_{total}/4)$ plot using Eq. (1):

$$\frac{1}{T_m} = \frac{R}{\Delta H°} \ln \frac{C_{total}}{4} + \frac{\Delta S°}{\Delta H°} \qquad (1)$$

where $R$ is the gas constant and $C_{total}$ is the total strand concentration of the oligonucleotides. For van't Hoff plots, $T_m$ values were determined for several oligonucleotide concentrations (2.43, 4.00, 6.60, and 10.9 μM). $\Delta G°^{37}$ was derived from $\Delta H°$ and $\Delta S°$ using Eq. (2).

$$G° = \Delta H° - T_m \Delta S° \qquad (2)$$

All experiments were performed in duplicate and repeated thrice, and the average results are shown.

### In-cuvette FRET assay

Fluorescence intensity was measured using a fluorometer (FP-8200, JASCO) by excitation at 555 nm before recording fluorescent counts with 590 nm emission filters. BHQ2-labeled mRNA mimics (100 nM, 45 μL) were added to equimolar amounts of 100 nM TAMRA-labeled hApo1nF or hApo1nF/PNA(C9) (45 μL), and the quenching rate of TAMRA fluorescence was measured. Based on a second-order reaction, $k_a$ was calculated from the slope of the fluorescence curve for the initial 30 s. All experiments were performed in duplicate and repeated thrice, and the average results are shown. TAMRA fluorescence was measured repeatedly ($n = 3$) using same samples.

### H&E staining and TUNEL assay

Dissected liver tissue was fixed by immersion in a 4% paraformaldehyde phosphate buffer solution for 24 h at room temperature. Paraffin-embedded tissue, H&E staining and TUNEL assay were performed by Kyoto Institute of Nutrition & Pathology, Inc. (Kyoto, Japan). TUNEL assay was carried out using in situ Apoptosis Detection Kit (TaKaRa). H&E staining was performed according to the protocol shown in Supplementary Table 7.

### Statistics & reproducibility

Statistical analyses were performed using GraphPad Prism9 software (Version 9.5.1). Significance was tested using ordinary one-way analysis of variance (ANOVA) (Figs. 2a, b, h–j, 3f, h, 4d, 6a, 7c, d) and two-way ANOVA (Figs. 2e, f, k, 6c, 7e) followed by Dunnett's multiple comparison tests. Dose dependence of knockdown activity (Fig. 2c) and hepatotoxicity (Fig. 2d), and drug concentration-dependent assessment of cell viability (Fig. 4c) were derived by nonlinear fitting using Graphpad Prism. No statistical method was used to predetermine sample size. No dataset was excluded, except for one replicate identified as an outlier by both the Grubbs' test (Alpha = 0.05) and the ROUT method (Q = 1%) using GraphPad software in Fig. 4d. All experiment findings were replicated at least three times. For all experiments, samples/animals were assigned randomly to experimental and control groups. Experiments were conducted without blinding, except for the followings: serum, RNA, and pathological samples were blinded and forwarded to the respective analysts, who performed measurements of AST/ALT, RNA-seq data acquisition, and pathological analysis.

### Reporting summary

Further information on research design is available in the Nature Portfolio Reporting Summary linked to this article.

## Data availability

All information regarding access to the primary and reference datasets from this study is provided in the main text, Supplementary Figures, and Tables. The RNA-seq data generated in this study have been deposited in the DDBJ Sequence Read Archive under DRA Accession "DRA016038" and are publicly available. Source data are provided with this paper.

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

## Acknowledgements

We thank H. Masumoto and H. Muto for technical support with RNA-seq, Y. Kayaba, and K. Nishimura for experimental assistance, H. Hisata for designing and manufacturing the synthesizer, and N. Horiuchi from Kyoto Institute of Nutrition & Pathology, Inc. for histological support. This study was conducted with the support of JSPS KAKENHI (22H02783 awarded to T.Y, B.C., K.Ohyama; 20H05874 to A.Y. and T.Y.; JP22KJ2503 and JP22J20853 to C.T.), the Center for Clinical and Translational Research of Kyushu University Hospital (A206 and A257 to T.Y.), the Cooperative Research Program of "Network Joint Research Center for Materials and Devices" (20221234, 20211123 and 20201156 to T.Y. and T.W.).

## Author contributions

T.Y. conceived the concept. T.Y., C.T., K.Ohyama, MA.S., T.W., M.HS. and A.Y. designed the study. C.T., K.Oh, R.T., N.A., and B.C. performed the experiments. All authors analyzed the data. T.Y. and C.T. composed the manuscript. All authors have read the manuscript and agree with its submission.

## Competing interests

At submission, T.Y. and M.H.S. were outside directors of Liid Pharmaceuticals. Post-submission, T.Y. and C.T. joined the company as a full-time director and an employee respectively, before acceptance. T.Y., C.T. and Liid Pharmaceuticals have submitted a patent application to the Japan Patent Office pertaining to the concept of BROTHERS technology of this work (International Application Number PCT/JP2022/036699). The remaining authors declare no competing interests.
