## [Peer Review File · Nature Communications]

REVIEWER COMMENTS

Reviewer #1 (Remarks to the Author):

The data presented in this paper seem well conducted and the impact of the complementary PNA strand on mediating either a nonspecific interaction of an LNA-phosphorothioate gapmer with protein or disrupting potential hybridization-dependent off-target RNA effects are plausible. This review is on the revised version of the paper which is substantially improved from the original. However, there are a few items that still need to be addressed. The results do potentially contribute to the overall knowledge of the field, but these details need attention for a high end journal such as Nature Communications.

Major Comments:

- The main premise of this paper is to assess the impact on a PNA complementary strand on two distinct interactions that potentially lead to ASO-mediated Toxicity. One stems from the nonspecific interaction of the oligonucleotide with proteins. This can be mediated by a combination of nucleotide sequence or phosphorothioate backbone interaction and are independent of hybridization. This source of toxicity is very different from those that might be mediated by a hybridization-dependent effect on unintended mRNA transcripts. The revisions in the paper make a very clear distinction between the impact of BRO on hybridization-independent v. -dependent effects. This is the most important advancement in this draft.
- The concept of 'toehold' being important in the effect on hybridization-mediated effects is hypothetical and not addressed by the design of the experiments. The explanation and data supporting the Toe-hold mechanism are now more clear in this version.
- In general the description of the results and discussion are now more clear. This reviewer still finds there are areas this could be further streamlined, and the discussion limited to the findings rather than getting into 'suggested' implication in some areas such as with the interpretation of the pathway mapping but this is a minor comment.
- It is acknowledged that efforts were made to address the cell uptake and distribution of the ssASO v. the BRO. In fact there was a demonstrable impact on the subcellular localization with a shift in the ration of cytoplasmic v. nuclear localization which was quite dramatic. I did not see the length of the BS mentioned in this experiment which should be included for interpretation of the results. Since the BRO context resulted in much less distribution to the nucleus the implication on activity should be addressed. This is why it is important to know what length of BS and the associated on-target activity was used. The question essentially is if this shift in distribution, which is attributed to differences in protein binding, occur without affecting antisense activity.

- P. 2/L. 61 – The examples given for Clinical programs being halted are not at all clear that they align with the premise of ASO-mediated toxicity. This was a minor comment before, but needs to be addressed. They are poor examples and the citations are press releases. This concept is central to the purpose of the paper and a more scholarly example should be used to avoid confusion.
- P. 3/L. 84 – There is still mention that the PNA duplex acts as a ‘repellent’ for protein binding. I previously commented there is no basis to support this statement, and if you insist on including it, the concept needs to be explained and defended with data. How does this “repell”. The concept implicates a change due to ‘like-charges’ repelling or a surfactant effect. Certainly there is an impact of BRO on binding relative to the ssASO but this is more likely either steric or electrostatic changes that affect the interactions.

Reviewer #3 (Remarks to the Author):

The Authors have resubmitted the manuscript to Nature Communications entitled "Dynamic and static control of the off-target interactions of antisense oligonucleotides using toehold chemistry". In doing so they have addressed several key comments made by the reviewer.

The authors have addressed the reviewer's concerns and corrected minor comments suggested. Thank you for the resubmission of Fig 5 it is now clear. They have also made significant effort to clarify sections with the addition of a reworded Fig 6.

The reviewer's only comment:

Major Comment 1: The reviewers addressed the question in the rebuttal.

However, the reviewer still feels it would be relevant to address this in the manuscript somewhere in the discussion - if not at least recognize this as a current limitation of the study. While the reviewer understands the authors may not have experience with other chemistries. But given the title "... off-target interactions of antisense oligonucleotides". The authors have only assessed one very specific chemistry - an LNA gapmer - to which no current FDA or EMA approved LNA gapmer ASOs exist. While approved therapeutic compounds are 2'MOE or 2'O-Me gapmer compounds. The last paragraph of the rebuttal should be included to infer their physicochemical principle in the manuscript.

In the following letter, we address the primary concerns raised by the reviewers. Comments from the reviewers are presented in italic, while our responses are highlighted in blue. Sections of the manuscript that have been revised are also marked in blue. We believe that these clarifications and adjustments will adequately address the feedback.

Reviewer #1

• It is acknowledged that efforts were made to address the cell uptake and distribution of the ssASO v. the BRO. In fact there was a demonstrable impact on the subcellular localization with a shift in the ration of cytoplasmic v. nuclear localization which was quite dramatic. I did not see the length of the BS mentioned in this experiment which should be included for interpretation of the results. Since the BRO context resulted in much less distribution to the nucleus the implication on activity should be addressed. This is why it is important to know what length of BS and the associated on-target activity was used. The question essentially is if this shift in distribution, which is attributed to differences in protein binding, occur without affecting antisense activity.

We sincerely appreciate Reviewer 1 meticulous review of our data. As Reviewer 1 observed from Fig. 5e, the ssASO predominantly localizes within specific organelles of the nucleus, discernible as regions of spotty absence of the blue Hoechst staining, likely representing nucleoli. Conversely, the BRO manifests a more expansive distribution across the nucleus, barring these nucleolar regions, in addition to the cytoplasm, underscoring a distinct disparity in their intracellular distribution patterns.

The mPCS1 ssASO used in this evaluation displays a sequence divergence of more than three bases from human PCSK9 mRNA (Page 4, line 113-114). This homology constraint precludes us from evaluating its knockdown efficacy within this human cell line context; nevertheless, the activity of the mPCS1/PNA(C8) BRO optimized in Fig.1 was comparable to that of its mPCS1 ssASO in mice (Fig. 1c). At the same time, as for toxicity, while the mPCS1 ssASO manifests discernible toxicity in both murine models and human cell lines, the BRO conspicuously obviates such toxicities (Fig. 1d and Fig. 5bc).

Citing a work by Liang et al., which has shown the possibility of ssASOs operating in both cytosolic and nuclear realms (Mol Ther. 2017 Sep 6;25(9):2075-2092), we postulate that our observed intracellular distribution variance in Fig. 5e may predominantly influence toxicity rather than therapeutic activity. This narrative aligns with the widely recognized model proposed by ST Crook et al. from Ionis, which

implicates the ssASO-protein interactions lead to disruption of cellular homeostasis (J Biol Chem. 2021, 296, 100416 etc.). However, at this moment, our dataset does not conclusively indicate whether any small number of predominant players dictates these intracellular distribution shifts or is a result of an aggregate influence of multiple factors as alluded to in Fig. 5abd. We aspire to delve deeper into the fundamental mechanisms responsible for toxicity employing BRO as a novel tool.

We have judiciously incorporated these insights into the manuscript without venturing into undue speculation. Additionally, for clarity, details about the length of the BS, initially outlined only in the caption of Fig. 5, have now been seamlessly incorporated into the main body of the text as well.

-Page 9, Line 302: The following phrase “mPCS1/PNA(C8) BRO optimized in Fig. 1.” was added.

-Page 13, Line 453: The following paragraph was added to the Discussion section. “The **mPCS1f** ssASO and the **mPCS1f/PNA(C8) BRO** used in the cellular distribution study in Fig. 5e display a sequence divergence of more than three bases from human PCSK9 mRNA, and this homology constraint precludes us from evaluating their knockdown efficacy within this human cell line context; nevertheless, the activity of the BRO was comparable to that of the ssASO in mice (Fig. 1c). At the same time, while the **mPCS1** ssASO manifests discernible toxicity in both mice and human cell lines, the BRO largely obviates such toxicities (Fig. 1d and Fig. 5bc). Therefore, citing a work by Liang et al.,⁶⁰ which has shown the possibility of ssASOs operating in both the cytoplasm and the nucleus, we postulate that the observed intracellular distribution variance between **mPCS1** ssASO and its BRO counterpart may predominantly influence toxicity rather than therapeutic activity.

This narrative aligns with the widely recognized mechanism proposed by Crook et al.,⁶¹ which implicates the ssASO-protein interactions lead to disruption of cellular homeostasis. In this context, a possibility for nucleolar transport of ssASO may be due to the previously-observed interaction of ssASO with paraspeckle proteins such as P54nrb (also known as Nono), PSF, and FUS.¹³ However, this study did not observe the apparent co-localization of P54nrb and **mPCS1f** ssASO in the nuclei (**Supplementary Fig. 5**). Of note, this study differs from previous studies as ASOs were introduced into cells using the CEM method, rather than conventional lipofection.⁶² These observations raise the question of whether any other specific factors, such as RNAs and proteins listed in Supplementary Table 2, dictates these intracellular distribution shifts.⁶³ We

aspire to delve deeper into the fundamental mechanisms responsible for toxicity employing BRO as a novel tool.“

- *P. 2/L. 61 – The examples given for Clinical programs being halted are not at all clear that they align with the premise of ASO-mediated toxicity. This was a minor comment before, but needs to be addressed. They are poor examples and the citations are press releases. This concept is central to the purpose of the paper and a more scholarly example should be used to avoid confusion.*

Thank you for the comment. We fully understand the intent of Reviewer 1. Due to the lack of scientific literature on these clinical compounds, we have substantially revised the paragraph in request as follows. We kindly ask for your review.

Page 2, Line 60:” However, although certain beneficial modifications can be identified to mitigate the off-target toxicity inherent to the parent ASO, these modification approaches often yield limited impact or can even exacerbate the toxicity profile.^{19,21,28} To date, these proposed methodologies rely on a trial-and-error basis. A more scientifically grounded approach, aimed at concurrently and effectively curtailing nonspecific interactions between RNA and proteins, may be preferred in addressing these toxicity challenges. Yet, to the best of our knowledge, no such strategy have been proposed.^{29,30}”

- *P. 3/L. 84 – There is still mention that the PNA duplex acts as a ‘repellent’ for protein binding. I previously commented there is no basis to support this statement, and if you insist on including it, the concept needs to be explained and defended with data. How does this “repell”. The concept implicates a change due to ‘like-charges’ repelling or a surfactant effect. Certainly there is an impact of BRO on binding relative to the ssASO but this is more likely either steric or electrostatic changes that affect the interactions.*

It seems that there is a discrepancy in the understanding of the term "repellent" between Reviewer 1 and the Authors. To avoid any potential risk of confusion, we have replaced it with the more explicit term "reduced". Please confirm.

-In Abstract: “These non-canonical ASO/PNA hybrids have been shown to have reduced non-specific protein-binding capacity.”

-Page 3, Line 85: “Therefore, the non-canonical ASO/PNA hybrids are expected to have reduced protein-binding capacity with ideal thermodynamic stability (Fig. 1c, Mechanism 1).”

Reviewer #3

Major Comment 1: The reviewers addressed the question in the rebuttal.

However, the reviewer still feels it would be relevant to address this in the manuscript somewhere in the discussion - if not at least recognize this as a current limitation of the study. While the reviewer understands the authors may not have experience with other chemistries. But given the title "... off-target interactions of antisense oligonucleotides". The authors have only assessed one very specific chemistry - an LNA gapmer - to which no current FDA or EMA approved LNA gapmer ASOs exist. While approved therapeutic compounds are 2'MOE or 2'O-Me gapmer compounds. The last paragraph of the rebuttal should be included to infer their physicochemical principle in the manuscript.

As pointed out by Reviewer 3, our study did not sufficiently evaluate ASOs incorporated with RNA analogs other than LNA. Therefore, we have explicitly added this point as a limitation of our paper. On the other hand, as noted, theoretically, similar strategies could potentially be applied to other types of ASOs, and we have included this consideration in our discussion. We have incorporated the following content into the 'Discussion' section of the manuscript.

-Page 15, Line 551, “One caveat of our present investigation is its predominant emphasis on LNA gapmers, thereby leaving room for the exploration of the applicability of the BRO strategy for ASOs with other RNA chemistries, such as MOE or 2'OMe, which have earned therapeutic approvals. Nevertheless, as a first approximation, the physicochemical principles presented herein are expected to possess considerable universality. This infers that the BRO architecture could feasibly be adapted to gapmers or splice-switching ASOs based on RNA analogs distinct from LNA, with the potential to confer comparable benefits. However, since the duplex-stabilizing ability of these RNA analogs is smaller compared to that of LNA, distinct optimal BS attributes (length/position/orientation) might be exhibited from those of LNA gapmers. Further detailed exploration tailored to diverse chemistries and modalities is anticipated in future research.

REVIEWERS' COMMENTS

Reviewer #1 (Remarks to the Author):

The authors have sufficiently responded to the 2nd round of comments to agree to publishing the paper.